# Direct printing of functional 3D objects using polymerization-induced phase separation

Bhavana Deore [1✉], Kathleen L. Sampson [1], Thomas Lacelle [1], Nathan Kredentser[1], Jacques Lefebvre [1], Luke Steven Young[1], Joseph Hyland [2], Rony E. Amaya [2], Jamshid Tanha [3], Patrick R. L. Malenfant [1], Hendrick W. de Haan[4✉] & Chantal Paquet [1✉]

3D printing has enabled materials, geometries and functional properties to be combined in unique ways otherwise unattainable via traditional manufacturing techniques, yet its adoption as a mainstream manufacturing platform for functional objects is hindered by the physical challenges in printing multiple materials. Vat polymerization offers a polymer chemistry-based approach to generating smart objects, in which phase separation is used to control the spatial positioning of materials and thus at once, achieve desirable morphological and functional properties of final 3D printed objects. This study demonstrates how the spatial distribution of different material phases can be modulated by controlling the kinetics of gelation, cross-linking density and material diffusivity through the judicious selection of photoresin components. A continuum of morphologies, ranging from functional coatings, gradients and composites are generated, enabling the fabrication of 3D piezoresistive sensors, 5G antennas and antimicrobial objects and thus illustrating a promising way forward in the integration of dissimilar materials in 3D printing of smart or functional parts.

---

[1] Security and Disruptive Technologies, National Research Council Canada, 100 Sussex Drive, Ottawa, ON K1A 0R6, Canada. [2] Department of Electronics, Carleton University, 1125 Colonel by Drive, Ottawa, ON K1S 5B6, Canada. [3] Human Health Therapeutics Research Centre, National Research Council Canada, 100 Sussex Drive, Ottawa, ON K1A 0R6, Canada. [4] Faculty of Science, University of Ontario Institute of Technology, Oshawa, ON L1H 7K4, Canada. ✉email: Bhavana.Deore@nrc-cnrc.gc.ca; hendrick.dehaan@ontariotechu.net; Chantal.Paquet@nrc-cnrc.gc.ca

Strategies to seamlessly integrate multiple materials into objects using 3D printing will enable the generation of new or improved properties and advance 3D printing as a mainstream approach to manufacture functional and smart objects[1–14]. Using reactive precursors, vat polymerization 3D printing provides a unique opportunity to spatially control materials from the surface to deep within the object[3,4,8,10,15–17]. For instance, the spatial, temporal, chromatic and intensity characteristics of light have been used in vat polymerization to pattern materials. Elegant examples include 3D printing using two wavelengths and orthogonal chemistries to spatially control two distinct polymerizations[9], light intensity and oxygen inhibition to modulate the crosslinking density[17] and photochromic molecules in combination with two wavelengths resulting in bioinspired materials with soft and hard sections[18]. Moore and Barbera have recently demonstrated the demixing of precursors to yield bicontinuous phases of polymer and pre-ceramic compounds with domain sizes controlled by light intensity[8]. Light-based printing techniques have also been used to photo-reduce in situ silver precursors yielding silver nanoparticles during the printing process[19–23].

Here, we demonstrate that by using purposely formulated resins, material phases within objects can be controlled using vat polymerization. The method utilizes polymerization-induced phase separation (PIPS), a process previously used to generate 2D patterns in holographic polymerization[24–29]. Exploiting concomitant changes in the thermodynamics of mixing that occur during polymerization, as well as spatio-temporal variations in monomer to polymer conversion, materials can be spatially directed towards the surface of the 3D printed object. The flux of functional material towards the surface of the printed object is controlled by balancing the kinetics of gelation, crosslinking density and rates of diffusion of the resin components. This approach has the benefit of generating material domains on the nanoscale and, thus, provides a means to combine macro-scale and micron-scale 3D designs with nanoscale material phases, features not easily achieved with nanoscale printing approaches such as two photon polymerization[30], localized electroplating[31,32], or metal ion reduction[33–35].

This report explores how resin formulation influences PIPS in vat polymerization (3D PIPS) and provides the insight needed to control material placement in printed objects. The use of 3D PIPS to spatially control material phases within printed objects opens up new opportunities to create functional coatings directly from printing or to generate composition gradients that are essential to reduce stresses that can manifest when integrating dissimilar materials[36]. We demonstrate the utility of the approach by producing conductive metallic silver features, enabling the fabrication of a dipole antenna array, strain sensors, as well as objects with antibacterial surfaces. Using the principles described herein, freedom to design material complexity/functionality directly into 3D printed objects can be envisioned to generate optimized catalytic supports, improve the wettability of biocompatible resins with hydroxyapatite particles, or embed anti-viral agents to minimize the transmission of pathogenic agents and will pave the way to new technologies in structural electronics[37], shape responsive parts for soft robotics, as well as smart objects with embedded sensors for the Internet of Things and wearables[13].

## Results

**Crosslinkers drive the spatial distribution of silver**. Here, we showcase a range of material morphologies that can be generated using photoresins containing a silver precursor as the non-polymerizable functional component (Fig. 1). The silver precursor, a mixture of silver neodecanoate (AgND) with 2-ethyl-2-

oxazoline, is ideal for this application as its molecular nature ensures higher diffusivity than larger functional materials such as nano-particles or micro-particles. Furthermore, because AgND does not scatter light as particles do, resins containing high concentrations of the complex can be printed. As has previously been shown with screen printable inks derived from this salt, the precursor will decompose into volatile products and conductive metallic silver traces with volume resistivity values as low as 9 μΩ•cm through a simple post printing sintering step using temperatures greater than 150 °C[38]. The resin systems studied herein were comprised of varying concentrations of polyethylene glycol (PEG) diacrylate crosslinkers and a monomer, 2-ethylhexyl acrylate (EHA, Supplementary Table 1). The four resin systems, distinguished by the length of the PEG spacer of the diacrylates, formed the basis of this study (170, 250, 575, and 700 g/mol $M_n$ PEG-diacrylates are referred to as DA-170, DA-250, DA-575, and DA-700, respectively; Supplementary Table 2). A concentration of 25 wt. % silver precursor (AgND) was used throughout the study based on results showing optimal electrical conductance at this concentration (Supplementary Table 3). A threshold concentration of ~19 wt. % AgND was required for electrical conduction using the 35 wt. % DA-250 formulation. The resistance of the surface of the cylinders did not change significantly when the concentration of AgND in the resin was between 25 and 38 wt. %, while AgND concentrations greater than 38 wt. % reduced the printability of the resin and resulted in brittle objects and less uniform silver coatings (Supplementary Fig. 1).

By adjusting the resin composition, it is possible to tune the morphology of the printed part from one where silver is concentrated at the surface forming a distinct coating to one in which the silver is dispersed throughout the object. A high concentration of silver at the surface of the printed object necessitates that the AgND migrates to the surface before becoming entrapped in the polymer network; this occurs when the kinetics of gelation are slow, and when the diffusion of AgND is not inhibited by the formation of a tight polymer network. A composite morphology, where the concentration of silver varies minimally throughout the 3D printed object occurs when the kinetics of gelation are fast and the AgND is impeded from migrating as a result of a tight polymer network. By dialing-in conditions with intermediate rates of gelation and crosslinking densities, gradients in silver concentration can be achieved as the AgND diffuses controllably away from the locus of polymerization and towards the surface of the object.

3D PIPS was first demonstrated by printing cylinders 1.5 mm in diameter and 2 cm in length and sintered at 210 °C to convert AgND into metallic silver. These resins contained 25 wt. % AgND yielding objects with 9.5 wt. % silver post-sintering (see Supplementary Fig. 2 and Supplementary Table 3). The morphologies formed by the various resins, both pre-sintering and post-sintering, can be seen in the SEM images taken at the edge of the cross-sections of the cylinders (Fig. 2a, b and Supplementary Fig. 3). The coating of the pre-sintered 25 wt. % DA-250 and DA-170 cylinders confirm that PIPS occurs during printing. The images reveal that silver accumulates towards the surface of the object with some resins forming a defined silver layer or coating (e.g., 25 wt. % DA-170, for details see Supplementary Table 2) while others produce a graded composition in silver (e.g., 99 wt. % DA-700, for details see Supplementary Table 2). These morphologies were assessed by performing 15 μm line scans at the edge of the cross-sections of the cylinders using Energy Dispersive X-ray Spectroscopy (EDS, Fig. 2c). With the exception of two of the DA-170 resins (50 wt. % and 99 wt. % crosslinker), all cross-sections show that the concentration of silver increases from the core to the surface of the cylinder (Supplementary Fig. 4). All four series have similar

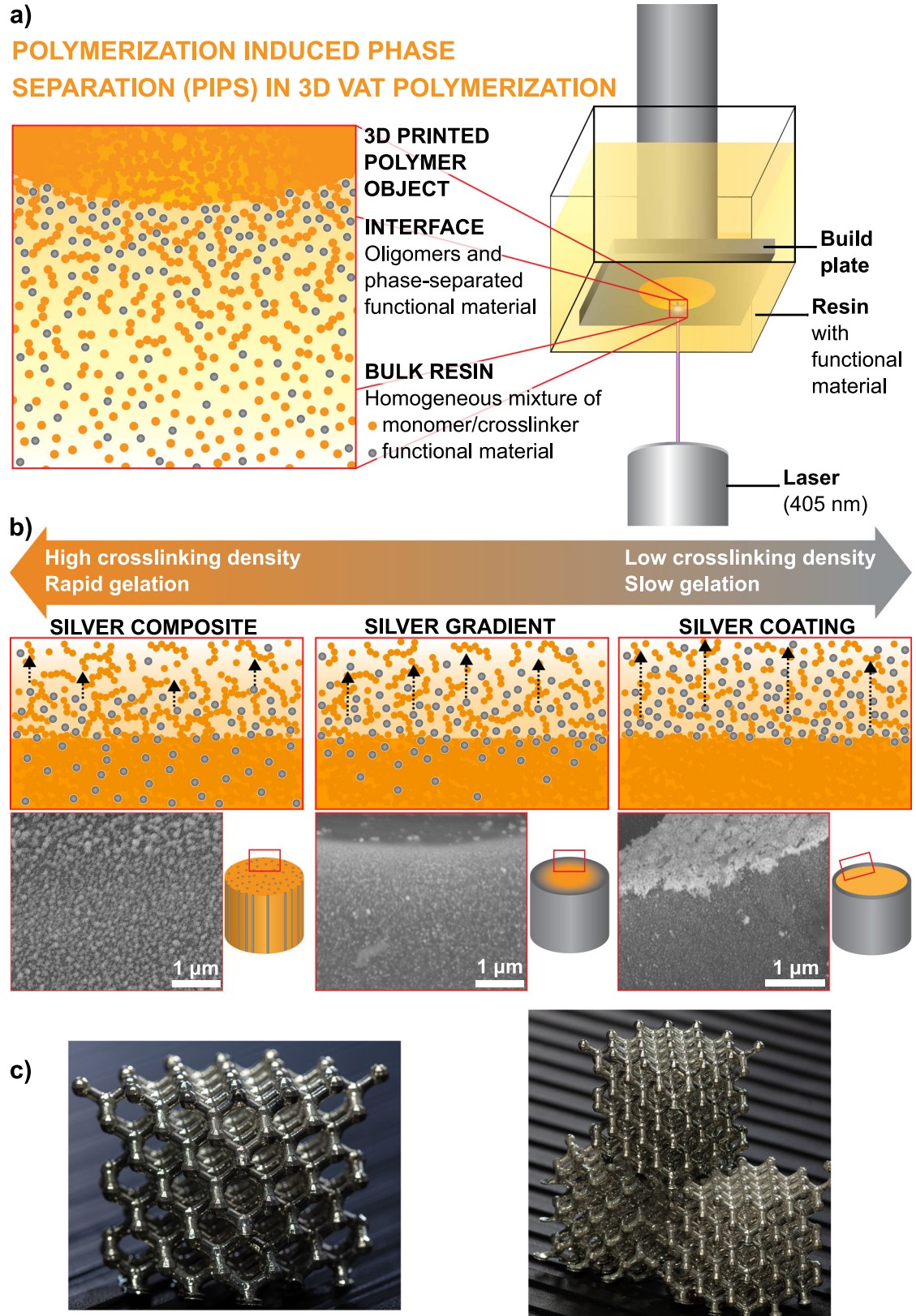

**Fig. 1 3D PIPS printing mechanism of functional objects. a** Schematic of the phase separation of AgND induced by PIPS (left) depicting the AgND functional material phase separating from the polymer object to the interface during 3D printing using an SLA printer (right). **b** The tuning of phase separation is controlled with crosslinker molecular weight and percent crosslinker to alter the crosslinking density and polymer gelation rate to yield different morphologies in 3D objects. Representative SEM images are presented to depict the composite, gradient or coating morphologies. **c** Photographs of PIPS induced 3D printed and sintered Ag coated lattice objects with dimensions of 40 × 40 × 40 mm.

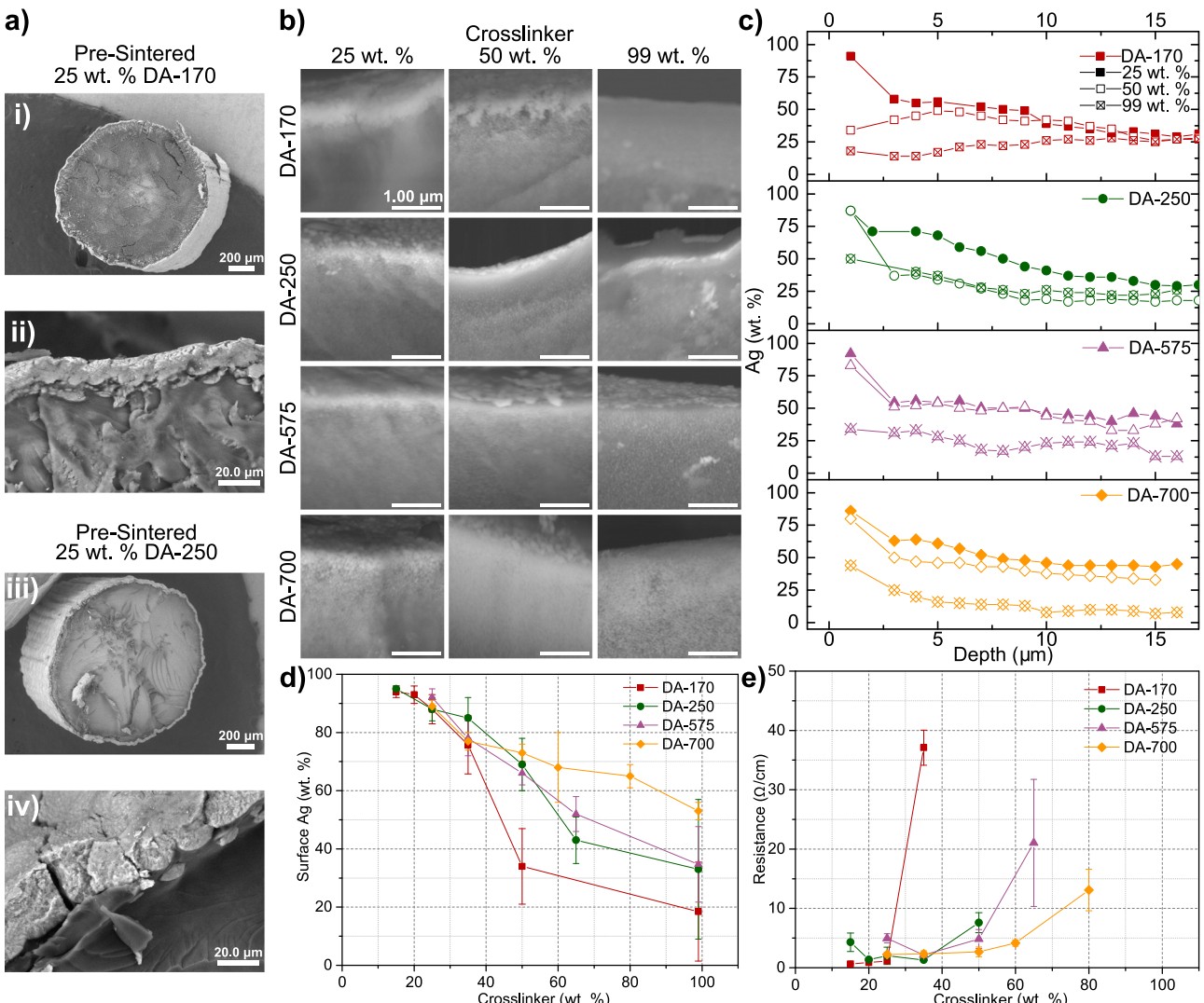

**Fig. 2 Silver phase separation as a function of crosslinking density.** Cylinders 1.5 mm in diameter and 2 cm in length were printed using resins containing 25 wt. % AgND. **a** SEM cross-sectional images of cylinders printed using resin mixtures (i, ii) 25 wt. % DA-170 and (iii, iv) 25 wt. % DA-250 and taken prior to thermal sintering, but treated with 5 min of UV curing to convert some of the silver salt to silver metal. These results confirm that the silver complex diffuses to the surface during printing and the partitioning of material phases does not happen post-printing. The cylinders once sintered at 210 °C contain 9.5 wt. % Ag. **b** Cross-sectional SEM images focused approximately <5 μm from the edge of the cylinder. Cylinders were printed using 25, 50 and 99 wt. % for DA-170, DA-250, DA-575, and DA-700 crosslinkers. **c** Weight fraction of silver as a function of depth with respect to the surface as measured by performing EDS on the surface of the object (data point at 1 μm) and on the cross-sections of the cylinders for 25, 50, and 99 wt. % crosslinker of each DA type. **d** Surface silver of sintered cylinders as a function of wt. % crosslinker as measured by EDS analysis on the surface of the object. The estimated interaction volume of the EDS beam for these measurements performed on the top surface of the object is ~2 μm. **e** Resistance as a function of wt. % crosslinker for post-sintered cylinders.

behaviors; the lower the crosslinker concentration, the more the silver concentrates at the surface to form a silver coating. However, resins made with diacrylates with long PEG segments show more of a graded distribution in the silver at the surface. To more easily compare among resin systems, the surfaces of the cylinders were analyzed by EDS to give the wt. % Ag within the first ~2 μm of the object as shown in Fig. 2d. These results illustrate that for all resin systems, the amount of Ag that accumulates at the surface decreases with increasing concentration of crosslinker, in agreement with the analysis of the cross-sections of the cylinders. These results also show that the short diacrylates yield structures with a broader concentration range of surface silver than resins made with the long diacrylates. For instance, the Ag surface concentration varies from 88 to 18 % for the DA-170 system, but only 86 to 40 % for the DA-700 system

when the crosslinker concentration increases from 25 to 99 wt. %. These results demonstrate that the spatial distribution of silver in the printed object is dictated by the length of the diacrylate crosslinker and its concentration.

In most printed samples, the silver concentration at the surface is sufficient to form a conductive film once sintered. The electrical resistances of the cylinder as a function of wt. % crosslinker of the resin (Fig. 2e) shows that for all systems, the resistances increase with increasing fraction of crosslinker, in agreement with Fig. 2d showing decreasing surface silver with increasing wt. % crosslinker. When the crosslinker concentrations are low, silver forms a coating with low resistance, owing to the high concentration of silver at the surface. As the crosslinker fraction increases, the coating progressively contains less silver, thus, increasing its electrical resistance. Above a certain fraction of crosslinker, the

surface silver is below its percolation threshold resulting in no detectable electrical conductivity. The relative change in the resistance with increasing crosslinker lengths also agrees with the trend in surface silver; the resistance of cylinders made with short diacrylates increases more dramatically than with the longer diacrylates in concurrence with the more significant decrease in surface silver for the short diacrylates. These results demonstrate that 3D PIPS is a simple, single-step method to generate functional coatings on 3D objects and, thus, circumvents the disadvantages of two-step coating methodologies such as poor film adhesion and uniformity (see Supplementary Fig. 5 for comparison). The calculated sheet resistance of 340 mΩ/sq is commensurate with values reported by Kell et al. in which screen printed traces using the same silver precursor have sheet resistance values of ~200 mΩ/sq. The values are three orders of magnitude lower than recently developed 3D printable conjugated polymers with values of ~6.6 × 10^5 mΩ/sq[39].

**Gelation rate, crosslinking density and diffusivity**. With the aim to resolve differences in the behaviors of the various resin systems and to develop a predictive model for 3D PIPS, we examined how the crosslinker influences the diffusion of the AgND during phase separation. Diffusion of phase separating components, such as AgND, is influenced by the rate a homogeneous resin mixture is transformed into an insoluble gel. This rate determines whether the AgND becomes trapped by the network or diffuses freely towards the unreacted resin where mixing is more favorable due to entropic gains. We measured the time required for a resin to form a gel, or delay time ($t_d$), by detecting changes in the refractive index when a resin converts from monomer to a polymer network using phase contrast optical microscopy (Fig. 3a and see Supplementary Movies 1–3). The $t_d$ values for the resin systems containing no AgND (see Supplementary Fig. 6 for resins with AgND) are summarized in Fig. 3b and, as expected, $t_d$ decreases with increasing wt. % crosslinker. For instance, a resin containing 15 wt. % DA-170 possesses an average $t_d$ of 5.8 s, while a resin composed of 99 wt. % DA-170 has an average $t_d$ of 1.8 s. Moreover, for a given wt. % crosslinker, the delay times decrease with increasing molecular weight (MW) of the crosslinker. This behavior likely results from the free end of long crosslinkers extending further from the polymer backbone, thus, increasing the probability of finding an unreacted acrylate group[40]. Fig. 3c shows how the delay times correlate to wt. % surface Ag; the longer the delay time, the greater the amount of surface silver. Therefore, resins that remain homogeneous mixtures of polymer, monomer and crosslinker for a longer duration afford more time of unimpeded migration for the AgND to reach the surface of the object.

Although Fig. 3c highlights how the delay time affects surface morphology, the results reveal that the resin systems generate different amounts of surface Ag for a given delay time, indicating that the amount of Ag that reaches the surface is not solely dictated by gelation rates. The effect is particularly pronounced at low delay times (i.e., high crosslinker concentrations) where resins made with long diacrylates yield objects with higher concentrations of surface silver than shorter diacrylates. We considered the role of miscibility between the AgND and the resin by comparing their calculated solubility parameters, δ (see Supplementary Table 1 and Supplementary Fig. 7). However, for a given wt. % crosslinker, the differences in solubility parameters for the different systems is marginal and do not explain the behavior highlighted in Fig. 3c.

The diffusivity of AgND will impact the amount of Ag that accumulates at the surface and may explain the observed differences in surface silver for a given delay time. The diffusivity

of AgND will change during polymerization as a result of increases in viscosity and constraints imparted by the growing polymer network[41]. The extent to which the diffusivity of AgND will change when the resin is transformed into a polymer network will be dependent on the length of the spacer between reactive moieties in the crosslinker or, in other words, the crosslinking density. To explore this idea, coarse-grained Langevin dynamics simulations of a simplified system were performed (see "Methods" section for details). The simulations tracked the displacement of a probe molecule, representing silver neodecanoate, in systems containing 100 wt. % crosslinkers. The simulations monitored the diffusivity of the probe molecule in unreacted resin and in a fully formed polymer network only, and therefore, the simulations did not require a consideration of kinetic effects of the polymerization reaction.

The diffusion coefficient of the probe molecule in crosslinkers with different linear bridging segment lengths (L = 3, 6, and 9) is shown in Fig. 4a. In the absence of any polymerization, the diffusion coefficient is higher in the short crosslinker. This is to be expected as the viscosity of the solution increases with increasing MW. However, for the case of diffusion in the polymer networks, the diffusion coefficient is highest for the longest crosslinker. In this limit, the networks formed by the shorter crosslinkers have a higher density of crosslinking points and, correspondingly, a tighter network for the probe molecule to travel through than the longer crosslinkers. This can be seen by examining the images for the probe molecule in the L = 3 network (Fig. 4b) and in the L = 9 network (Fig. 4c). The length of the linear bridging segments (blue segments of Fig. 4b, c) and the density of the crosslinking points (red segments) that define the density of the polymer network are distinctly different in the L = 3 and L = 9 network. The shorter length between crosslinking points of the L = 3 network causes the diffusion of the probe molecule to be more constrained in comparison to the L = 9 network. Note that the y-axis in Fig. 4a is logarithmic and, thus, the decrease in diffusivity for the short crosslinkers is much more dramatic than for the long crosslinkers. The diffusion of the probe molecule is ~24 times greater in the unreacted resin than in the network for L = 3, but it is only ~2 times greater for L = 9.

The inset to Fig. 4a shows the diffusion coefficient of the probe molecule in the polymer networks as a function of crosslinker length. Of note, the diffusion coefficient for the probe molecule is larger in the L = 9 network than the L = 3 network indicating that the diffusivity of AgND is very likely to be lower in the polymer network made with a tight crosslinked network. Reduced diffusivity in the tight network formed by short crosslinkers will impede the ability of AgND to migrate to the surface and, thus, provides a rationale for the lower surface Ag found in the system with shorter crosslinkers at low delay times. These results show how the interplay between the rate of network formation and the temporal changes in diffusivity of AgND that occur during polymerization affect the extent to which AgND migrates to the surface, which ultimately dictate the spatial distribution of silver in the object.

**3D PIPS for smart objects**. Using purposefully formulated resins to control the placement of AgND, we demonstrate the value in being able to tune the surface morphology of printed objects with particular functional properties by considering three applications: strain sensors, antennas and antimicrobial objects. The 3D PIPS approach provides the ability to generate strain sensors that combine complex 3D geometries with piezoresistive properties and holds promise in wearable electronics and motion sensing[42]. Truss structures were 3D printed using a resin formulation that yields graded silver compositions. The silver is sufficiently

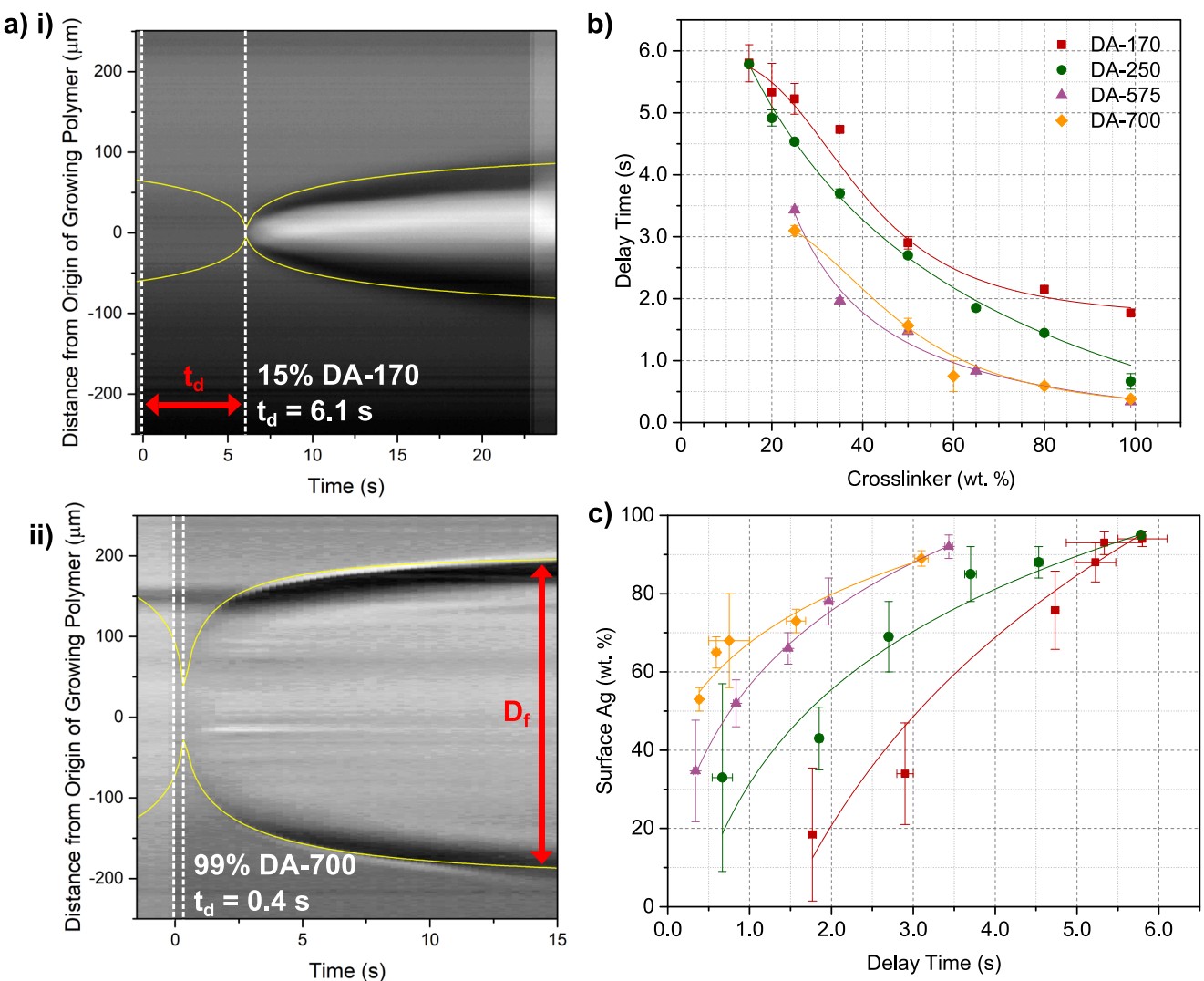

**Fig. 3 Dynamical study of photoresins by optical microscopy. a** Constructed images obtained from movies of islands of polymer formed when resin/ crosslinker inside a capillary is exposed to a focused 405 nm laser spot. A slice through the center of the island reveals its two opposite edges and is plotted as function of time. Two examples (i) and (ii) are shown for 15 wt. % DA-170 and 99 wt .% DA-700. **b** Delay time, $t_d$, extracted from a fit to Eq. 1 (See Experimental Section) as a function of wt. % crosslinker. **c** Weight % of Ag extracted from SEM/EDS analysis at the surface of the cylinder as a function of $t_d$ for various resin formulations. The delay times are from the same gelation experiments exposed to a 405 nm laser in part **a**, **b** for each formulation.

concentrated at the surface to form a percolated path for electrical conduction; however, polymer inclusions within the surface silver layer introduces barriers to conduction. With the printed objects being compressible (see Supplementary Fig. 8 for selected elastic modulus of cylinders), upon applying pressure, the polymer matrix deforms creating new contacts between the silver domains, increasing the conduction pathways and, thus, decreasing resistance (Fig. 5a). By controlling how the silver salt migrates, we can modulate the density of silver at the surface and, thus, its electrical response to compression. The SEM images of the surfaces and cross-sections of strain sensors of Fig. 5b, made using 39 and 58 wt. % crosslinker, demonstrate the differences in the amount of silver present at the surface. For the truss made with 39 wt. % crosslinker, the surface features a dense film of Ag nanoparticles with low electrical resistance whereas the truss made with 58 wt. % crosslinker has a surface morphology with sparser particles and, correspondingly, higher electrical resistance (Supplementary Fig. 9). The relative decrease in resistance depends on the extent to which the conduction pathway is hindered by the presence of polymer at the surface. Thus, trusses made with high crosslinker

concentrations have silver coatings with a more obstructed conduction pathway in comparison to trusses made with less crosslinker. Therefore, these trusses will create a greater number of new silver-to-silver contacts during compression resulting in a greater change in resistance. As shown in Fig. 5c, the change in resistance increases with the wt. % crosslinker used to make the trusses. The benefit of this approach is that as opposed to varying the filler loading to tune the sensor response, here 3D printed piezoresistive sensors can be made to respond with a given electrical response by simply controlling the phase separation of silver through the resin formulation. In addition, segregated silver at the interface allows one to reach a percolation threshold with a lower loading of conductive filler in comparison to commonly used conductive composite morphologies[43] making more efficient use of the conductive material, improving the electrical conductance of the silver phase and minimizing impact on the mechanical properties of the bulk object polymer phase. The strain sensors were found to have gauge factors (i.e., ratio of relative change in electrical resistance to the mechanical strain) of 2.3, 3.2, 5.1, and 15.7 for the sensors made with 39, 46, 52, and 58

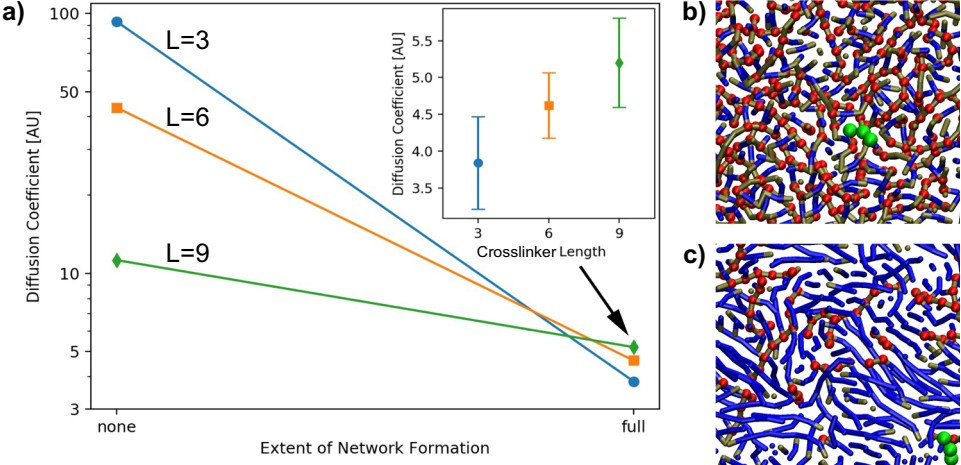

**Fig. 4 Simulation Result for Diffusion Coefficient. a** Diffusion coefficient of a probe molecule in crosslinkers of different spacer lengths (L = 3, 6, and 9) before and after the formation of a polymer network. The inset shows the diffusion coefficient of the probe molecule (i.e., a molecule 3 beads long), representing silver neodecanoate as a function of the crosslinker spacer length in the formed network. Snapshots from the simulations of the probe molecule (green molecule) in a network formed with crosslinker (**b**) L = 3 and (**c**) L = 9. The red balls represent crosslinking points while the blue parts are the bridging segments of the crosslinker. The high density of crosslinking points of the L = 3 system creates a tight polymer network that impedes diffusion of the probe molecule. In comparison, the L = 9 system allows for less constrained diffusion of the probe molecule due to its lower density of crosslinking points and longer segments between crosslinking nodes.

wt. % crosslinkers as shown in Fig. 5d, similar to those reported for 2D strain sensors[44]. This example illustrates how gauge factors can be dialed-in by simply varying the crosslinker concentration enabling one to target a sensitivity regime for 3D printed piezoresistive sensors.

3D printing is ideally suited to fabricate millimeter wave antennas for 5G as 5G will function on small networking cells that use arrays of antennas in small geographical areas requiring a large number of integrated low loss devices. These requirements can be achieved by using 3D printing to make antennas low cost, in arrays and embedded in objects. Moreover, by suspending the antenna in air using a 3D design, signal loss can be minimized with air becoming the effective dielectric. Using the 3D PIPS approach, we fabricated an array of 3D printed dipole antennas and demonstrated transmission of 2.4 GHz waves. The dipole antenna array, shown in Fig. 6a, displays the radiation pattern found in Fig. 6c as measured using an anechoic chamber (Fig. 6b) and its comparison with the theoretical response for a dipole array on a ground plane. Focusing of the radiation pattern into a main lobe is the result of radiation interference between antenna elements. The half power beam width of the theoretical pattern is 48° compared to 45° for the measured pattern, resulting in a remarkably small difference of 3° and, thus, demonstrating the suitability of this printing process for antenna applications. The gain measurement performed in an anechoic chamber using a gain standard horn antenna, as shown in Supplementary Fig. 10, are comparable with literature reports[45].

Antibacterial properties of nanoparticle silver have been used in many medical and dental applications for the prevention of infection[46–48]. To evaluate the antibacterial behaviors of 3D printed Ag objects, a halo inhibition zone test against *E. coli*, as well as bacterial growth kinetics were carried out along with control objects containing no silver. A concentration of 0.5 or 1.0 wt. % Ag was used in this study in order to form Ag nanoparticles rather than a film on the surface (Supplementary Fig. 11). As seen in Supplementary Fig. 12, the 3D objects containing Ag show a bacterial inhibition zone on an agar plate and bacterial inhibition in the liquid medium of *E. coli* while the control samples show growth of *E. coli*. These results demonstrate the antibacterial properties of 3D printed Ag objects and illustrate how 3D PIPS

could provide a means to embed small quantities of antimicrobial or antiviral agents at surfaces of printed objects helping to minimize the transmission of pathogenic bacteria and viruses.

The concept of 3D PIPS can be applied broadly to fabricate 3D objects with different surface properties by using various functional materials with resins that induce migration of these materials towards the surface. 3D PIPS was used to generate 3D objects with various surface composition, as illustrated in the examples of Supplementary Fig. 13 that use resins loaded with nanoparticles. Our 3D PIPS approach thus represents a powerful means to make functional coatings and functionally graded materials of various compositions.

In summary, we have showcased how the temporal and spatial variation in monomer-to-polymer conversion that takes place in vat 3D polymerization causes local demixing of functional materials, triggering diffusion of these materials towards the bulk resin. By harnessing the rate at which the functional materials become entrapped in the polymer network during 3D PIPS, a wide range of surface morphologies can be accessed. The insight gained in controlling the material phases allows a rational approach to formulating resins to access a wide range of material morphologies for specific applications. Due to the universality of this approach, 3D PIPS represents a powerful method to create materials with a continuum of morphologies using a vast material set and will accelerate the adoption of vat polymerization as a viable technique to generate functional 3D objects.

## Methods
### 3D printing functional photoresin preparation
*Photoresin preparation.* Components used to make photoresins are shown in Supplementary Table 1. Acrylate photoresins were prepared by varying the amount of monomer and crosslinker ranging from 15 wt. % to 99 wt. % crosslinker with 1 wt. % photoinitiator (ethyl (2,4,6-trimethylbenzoyl) phenylphosphinate, TPO-L) and the remaining wt. % monomer as shown in Supplementary Table 2. The combined mixture was vortex mixed for 30 s before use.

*Photoresin with AgND.* The silver precursor, a mixture of silver neodecanoate with 2-ethyl-2-oxazoline silver neodecanoate referred to as AgND, was prepared by mixing 2.5 g of silver neodecanoate in 0.54 g of 2-ethyl-2-oxazoline (1:0.22 weight ratio) using a planetary mixer at 2000 rpm for 4 min followed by 2200 rpm for 30 s resulting in a 82 wt. % of silver neodecanoate in 2-ethyl-2-oxazoline. To make resins loaded with silver precursor, 3.0 g of AgND was added to 9.2 g of photoresins

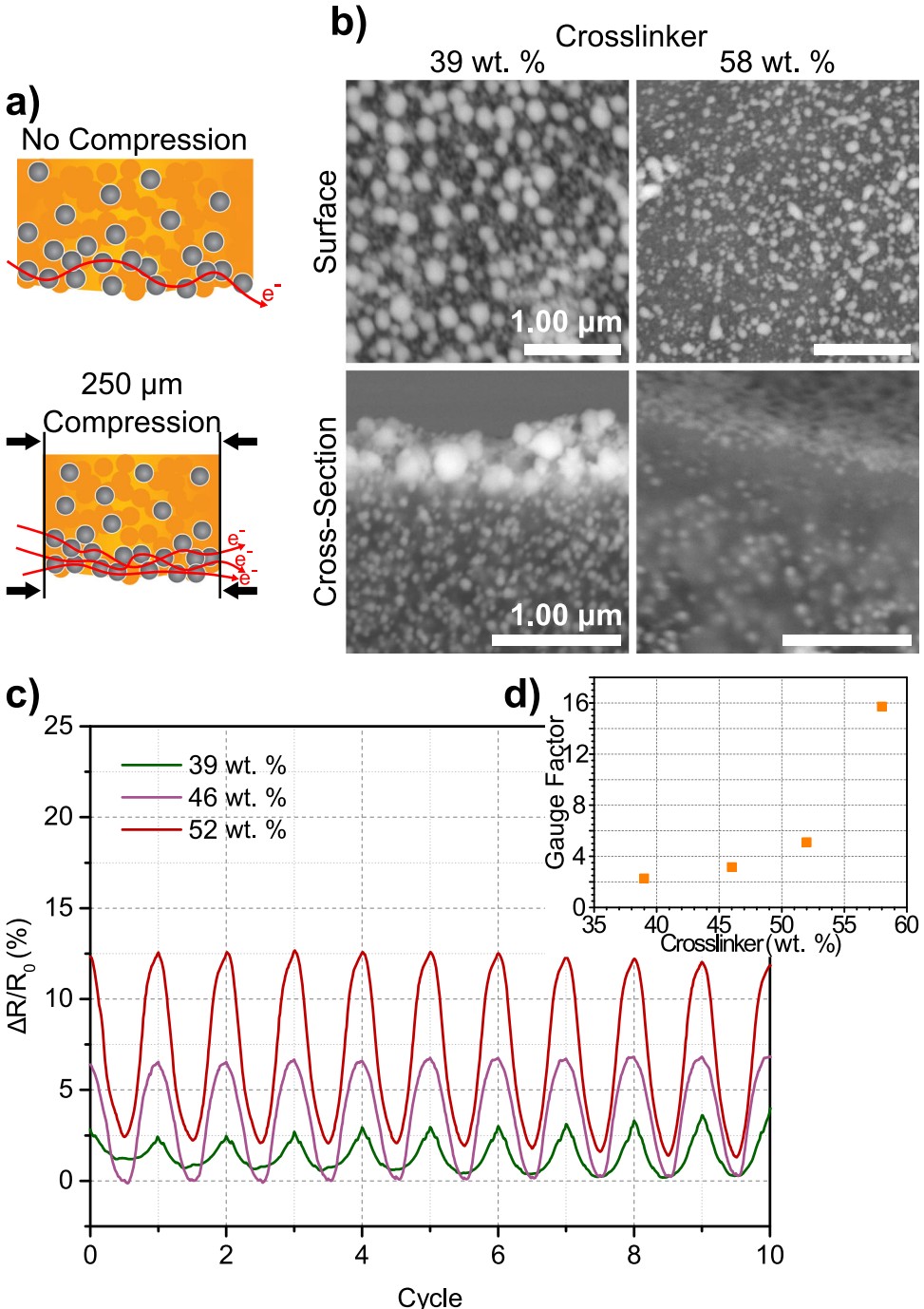

**Fig. 5 Strain sensor measurements. a** Schematic representation of the compression mechanism within a truss. The greater the wt. % of crosslinker used, the more the conduction channel is obstructed by the polymer phase. With an applied pressure, the polymer phase deforms creating new silver-to-silver connections and decreasing the resistance. The trusses that have highly impeded conduction channels due to a high polymer fraction at its surface create a higher number of new silver-to-silver connections upon compression than the truss with less polymer and, thus, respond accordingly with a greater decrease in resistance. **b** SEM images of the top surface and cross-section of truss structures prepared using different wt. % crosslinker. **c** Change in resistance upon compression cycling of truss structures prepared with different wt. % crosslinker. Trusses made with 39, 46, and 52 wt. % crosslinker responded to a compression of 250 µm with maximum changes in resistances of 2.5, 6.0, and 12.5%, respectively. **d** Gauge factors of the trusses as a function of the concentration of crosslinker used in the resin. The gauge factor increases with increasing crosslinker concentration illustrating the strength of this approach in being able to target a given piezoresistive sensor sensitivity through the formulation of the resin.

yielding compositions containing 25 wt. % of AgND. These resin formulations are comprised of 7.9 wt. % of silver metal or 9.5 wt. % silver metal post-sintering assuming neodecanoate and 2-ethyl-oxazoline become volatile during the process (see TGA in Supplementary Fig. 2). The amount of silver in the printed object in comparison to the calculated amount of silver in the formulation was confirmed using TGA of cylinders containing 25 wt. % AgND (or 9.5 wt. % silver post-

sintering) as shown in Supplementary Fig. 2a, b. A concentration of 25 wt. % AgND in the resin was found to give optimized resistances (see Supplementary Table 3).

*Resin preparation for strain sensor*. Mixed DA-resins were prepared by mixing different volumes of two separately prepared DA-resins such as 75 mL of DA-575

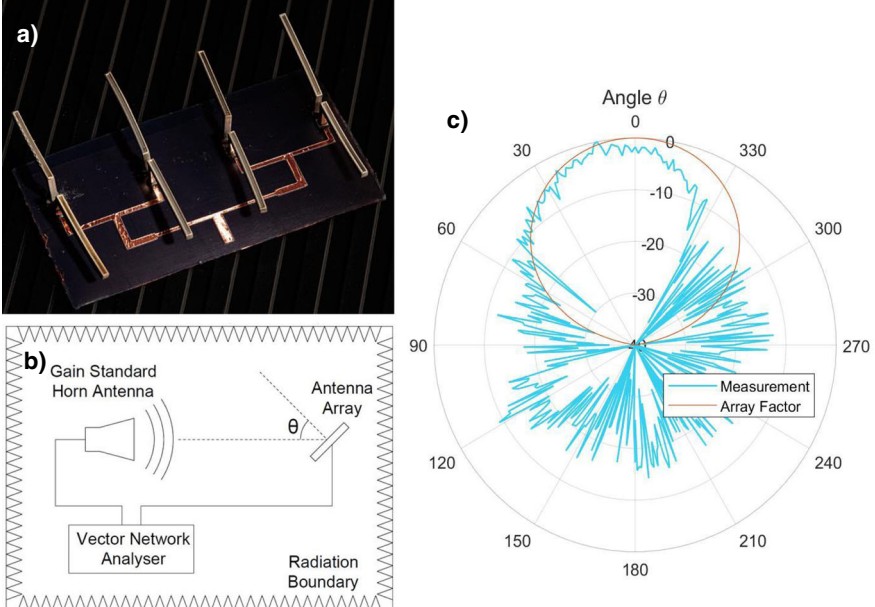

**Fig. 6 3D printed dipole antenna array. a** Photograph of antenna array (dimensions 10 cm by 10 cm). **b** Functional anechoic chamber layout showing transmission between a gain standard horn antenna and antenna array under test. **c** 3D printed dipole antenna array normalized radiation pattern at 2.4 GHz compared with ideal array factor.

resin (40–65 wt. % DA-575) with 25 mL of a 35 wt. % DA-250 resin to vary the total amount of crosslinker between 39–58 wt. %. To make functional resin, 3.0 g of Ag precursor was then added to 9.2 g of mixed DA-resin to adjust the amount of Ag metal in the resin to 7.9 wt. %. The combined mixture was then vortex mixed and used to print truss structures used as the strain sensors.

*Resin preparation for dipole antennas by phase separation.* Resin mixtures were prepared by mixing different volumes of two separately prepared DA-resins, namely, 75 mL of 50 wt. % DA-575 with 25 mL of 35 wt. % DA-250. 3.0 g of Ag precursor was then added to 9.2 g of mixed DA-resin to adjust the amount of Ag metal in the resin to 7.9 wt. %. The combined mixture was then vortex mixed and used to print dipole antennas.

*Preparation of dipole antenna by electroless silver plating.* The mixed DA-resins were prepared by mixing different volumes of two separately prepared DA-resins such as 75 mL of 50 wt. % DA-575 and 25 mL of 35 wt. % DA-250. The dipole antennas were printed using the silver-free resin and subsequently seeded with silver particles to yield an adherent and uniform silver coatings on the antenna substrate. For a seeding pre-treatment procedure, antennas were dip coated with 100× dilute solution of commercial nanoparticle ink (SunTronic™ NANOSILVER) in toluene. The thin layer of seeds did not result in any measurable conductivity; however, this step was crucial to obtain uniform and adherent silver coating through electroless plating. Further, electroless plating procedure was optimized by varying the rate of addition and concentration of the silver plating bath solutions in order to obtain coatings with low surface roughness[49]. The electroless silver plating bath was composed of solution A (0.2 g of glucose, 0.02 g of tartaric acid, and 0.5 mL of ethanol in 20 mL of deionized water) and solution B, a $Ag(NH_3)^{2+}$ solution (0.2 g of $AgNO_3$, 0.075 g of NaOH, and 0.5 mL of ammonia in 20 mL of deionized water). Over a period of 60 mins, under constant stirring, these two solutions were combined by dropwise addition of solution B into solution A in a bath containing the dipole antenna at room temperature. The antennas were then collected, rinsed with water and dried at 140 °C for 5 min. The adhesion of the silver coatings of the samples prepared with the optimized electroless plating conditions was compared to those obtained through phase separation (Supplementary Fig. 5). The adhesion of the silver coatings was determined by applying Scotch tape to the sample, pressing firmly and removing the tape.

*Functional resin preparation for antibacterial activity.* 3D objects for antibacterial activity measurements were prepared using 0.0 to 1.0 wt. % of Ag metal in 35 wt. % DA-575 resin.

### Printing, sintering and characterization of 3D objects
*SLA printing of 3D objects.* 3D objects using functional material were printed using Peopoly Moai Laser SLA 3D Printer (Technical Specifications: Build Volume: 130 × 130 × 180 mm, Laser spot size: 70 microns, Laser wave length: 405 nm, Laser

power: 150 mW (measured power output 29.2 mW), Machine size: 330 × 340 × 660 mm, Layer Height: 10–200 microns). Objects were printed using a fluorinated ethylene propylene liner-coated vat with laser power rating set to 75 and all other settings set to the default. All cylinders were printed using light with an intensity of 760 W/cm² other than optimization experiments where 350 and 560 W/cm² were used to explore the role of light intensity on the phase separation of silver as described in the Supplementary Information (Supplementary Table 4 and Supplementary Fig. 14).

*Sintering of printed 3D objects.* 3D objects were thermally sintered on a Kapton sheet at ~210 °C (substrate temperature) for an hour using a reflow oven under nitrogen with 500 ppm oxygen to convert silver neodecanoate to metallic silver. The mass loss of the 3D printed cylinders that contained no silver were analyzed by TGA under isothermal conditions at 210 °C. The mass loss after 1 h is minimal demonstrating that the polymer does not degrade significantly during the sintering step (See Supplementary Fig. 15).

*Characterization of 3D objects.* The resistance (R) values were taken by measuring on the walls of the cylinders with the probes separated by 1 cm. Lead contact resistance was found to be negligible, and therefore, a two probe measurement method was used. Average resistances were obtained by measuring 10 different cylinders. Sheet resistances were calculated based on the circumference of the cylinder (W) and the length (L) between electrical probes (1 cm), where the sheet resistance ($R_s$) is $R_s = R \times W/L$. Scanning electron microscopy (SEM) imaging and Electron Dispersive X-ray Spectroscopy (EDS) surface and cross-section analysis were performed with a Hitachi SU3500 using an acceleration voltage of 15 kV (SEM) and 30 kV (EDS) and a spot size of 30.

*Optical studies of resin (polymerization kinetics).* The photopolymerization was studied using an experimental method employing an optical microscope to observe the changes in refractive index which occurs as the resin polymerizes under illumination by a 405 nm laser as shown in Supplementary Fig. 16. Each resin was prepared in 2–10 mL quantities (see Supplementary Table 2 for formulations) and loaded into a capillary micro glass slide (0.10 × 2.0 mm, 0.10 mm wall thickness; Electron Microscopy Sciences). The resins were prepared without AgND as its presence made it more difficult to clearly determine the interface between the polymer island and remaining resin in the capillary tube, but a similar gelation time trend was found between resins with and without AgND for the DA-575 cross-linker (Supporting Information Fig. 6). The filled capillary micro glass slide was placed under the optical microscope (Nikon ME600) fitted with a 405 nm laser and optical filters (405 nm laser line and 405 nm Raman edge filters) to prevent saturation of the camera with excess laser light. Laser power (10 µW with 1–2 µm diameter spot size and ~566 W/cm² intensity) was set low enough to reveal polymerization in a 10–60 s timescale. Refractive index changes can be best visualized using phase-contrast imaging mode. Movies were acquired by a color camera (Luminera Infinity2) and include an initial phase without laser

illumination, followed by removal of a laser shutter where resin/crosslinker exposure is initiated. A small dot is observed after a few seconds which subsequently grows to an island several micrometers in diameter (200–400 µm). The shutter is closed and recording stopped once the island stopped growing (see Supplementary Movies 1–3). Each recorded Movie was subsequently analyzed with ImageJ. For a polymerized island, a slice through the center was selected to construct a plot of the island size as a function of time (Supplementary Movie 3). The start time was accurately determined from the slight change of illumination conditions when the shutter was On or Off. The two opposite edges of an island and their time evolution can be clearly identified and fitted to Eq. (1):

$$\text{Profile} = \pm \left( D_f - D_f e^{-\sqrt{|(t-t_d)|/t_c}} \right) \quad (1)$$

where $t_d$ is the delay time, $t_c$ is a rate dependent parameter, and $D_f$ is a size dependent parameter. The $t_d$, which represents the time elapsed between when the laser is turned on and the first observable sign of polymer network, serves as a measure of relative gelation time. This was repeated two more times with newly filled capillary tubes for each formulation measured.

**Coarse-grain modeling of the diffusivity of a probe molecule.** The simulations used standard coarse-grained (CG) polymer methodologies[50] to construct a system roughly modeled on the experimental setup. A cubic box was filled with CG polymers that represent the crosslinkers. Each polymer consists of L beads that are linearly joined together via finitely extensible nonlinear elastic (FENE) spring bonds[51] to prevent bond crossing. Intermolecular and intramolecular interactions between beads were implemented using the Weeks–Chandler–Andersen (WCA) potential[52] such that there is no attraction between beads, but instead there is only short ranged repulsion that yields excluded volume. Stiffness is imparted to each polymer via a harmonic angle bond that causes a linear alignment of any three consecutive monomers to be the energetically favorable conformation. A linear arrangement of 3 beads, which approximate the size silver neodecanoate, was used as a probe molecule. Each simulation included a single probe molecule.

A length of L = 3 beads was chosen to roughly correspond to the length of the DA-170 crosslinker. Silver neodecanoate, which is of similar length to DA-170, was also modeled as an L = 3 molecule. Simulations were also performed for crosslinker molecules of length L = 6 and L = 9 to study how the dynamics change with crosslinker length. The L = 9 molecule was the longest length that could be studied due to constraints on the simulation setup. Thus, these simulations do not replicate the crosslinker ratios studied experimentally and instead explore the dependence on crosslinker length in a more general way.

The diameter of each bead was set to σ and, thus, σ serves as the length scale for the simulation. The box length was set to 20 σ and each system was filled with enough polymers to achieve a volume fraction of 0.491. Simulations using the same particle model have demonstrated that the system will begin to crystallize at a volume fraction of 0.492[39]. Setting the volume fraction to 0.491, thus, mimics the experimental setup by retaining liquid-like behavior while ensuring that no voids will form during the polymerization process. The systems then consisted of: 2500 L = 3 crosslinkers +1 L = 3 probe molecules, 1250 L = 6 crosslinkers +1 L = 3 probe molecule, and 833 L = 9 crosslinkers +1 L =3 probe molecules. Images from the L = 3 and L = 9 systems are shown in the Supplementary Fig. 17.

For each crosslinker length, simulations were performed for two scenarios: diffusion of the probe molecules in pure resin (no polymerization of crosslinkers) and in the final network (saturated polymerization). For both scenarios, the system was evolved in time via Langevin dynamics[50] using the HOOMD blue simulation package[53,54].

The setup of each system consisted of a number of preliminary steps. First, the box was filled with the specified number of polymers and one probe molecule being constructed on a grid. Second, the system was randomized with a short simulation where there were no intermolecular interactions. This allows all molecules to pass through each other and, thus, randomizing the system very quickly. Third, the excluded volume interactions between beads on different molecules was ramped up until the full potential was applied. This method yielded the initial configurations shown in the Supplementary Fig. 17.

For the simulations of diffusion in pure resin, these initial states were then evolved in time and the position of the probe molecule was monitored. The mean square displacement (MSD) of the probe molecule was then calculated from these trajectories by internal averaging. The diffusion coefficient was extracted from a linear fit of the MSD plotted against simulation time.

For the simulations of diffusion in the network, simulations were performed in two steps. In the first step, a polymer network was formed by setting 10% of the crosslinkers to be reactive allowing polymerization to take place via diffusion. These network formation simulations were conducted for a long enough time period that the rate of adding new crosslinking points became very small. The resulting networks were then considered fully polymerized. Once the network was formed, a second simulation was conducted in which the diffusion of the L = 3 probe molecule representing silver neodecanoate was monitored. No further polymerization occurred during this step.

Analysis of the MSD and resulting diffusion coefficient was conducted in the same manner as the pure resin case. However, there are more sources of variability in the network case since the network is not homogenous (while the pure resin essentially is). This means that the rate of diffusion depends on the local environment and, thus, varies as the probe explores different areas of the network. Further, since the simulations are limited in the size of the network that is constructed, the results will also vary between different simulations. Even though the simulations are identical in procedure, if a different seed is used to initiate the dynamics then the network that is formed will be significantly different between simulations and, thus, the calculated diffusion coefficient may also be significantly different.

To account for these variations, simulations were performed using three different initial seeds to build three independent networks for each crosslinker length. The final diffusion coefficient given in the main manuscript is the average of the value calculated for each realization. The error bars correspond to the standard error as calculated across the ensemble of three.

## Applications

*Strain sensor measurements.* The 3D-printed truss objects (11.24 × 11.24 × 13.40 mm) were affixed to a programmable linear stage (Zaber Technologies; model X-LRQ150AP-E01) with aluminum tape and two-component silver epoxy (#8330S-21G, MG Chemicals) as shown in Supplementary Fig. 18. The stage was controlled using the LabVIEW software package and changes in electrical resistance were measured using a digital multimeter (Keithley Integra Series 2701 Ethernet Multimeter/Data Acquisition System) with probes connected to the aluminum tape. All samples were measured at a compression/elongation rate of 625 µm/s in increments of 5 µm with a 1 s pause before each resistance measurement. The gauge factor was calculated as follows:

$$\text{GF} = \frac{\left[ \frac{\Delta R}{R_0} \right]}{\left[ \frac{\Delta L}{L_0} \right]}$$

where $\Delta R$ is the difference in resistance between zero compression ($R_0$) and a compression of 250 µm (R), $L_0$ is the length of the truss (13.4 mm) and $\Delta L$ is the change in length (250 µm).

*Dipole antenna measurements.* Dipole antennas were 3D printed using functional resin with Ag precursor adjusted so that the amount of Ag metal in the resin was 7.9 wt. % and then glued on the substrate with conducting epoxy. A microstrip array was used to feed four dipole antennas, which were metalized after 3D printing via phase separation of Ag and post-printing sintering. The photograph of the antenna array is shown in Fig. 6a. The dipole antenna measurements were performed in an anechoic chamber (Fig. 6b). The antennas were designed to be centered at 2.4 GHz with a physical length of 6.25 cm. A gain standard horn antenna was positioned at one end of the chamber and connected to one port of a Vector Network Analyser (VNA) through an amplifier. The device under test (antenna array) was placed at the opposite end of the chamber on a rotating mount and connected to the other port of the VNA. While rotating the antenna array, s-parameter measurements were taken to determine the radiation of the antenna as a function of angle. Only the positive going half of the radiation pattern was used to determine the half power beam width. A signal present at angles between 90° and 270° were due to the finite limitation of the ground plane and the noise naturally present in the system.

*Antibacterial activity: Halo inhibition zone test.* Bacterial suspension TG1 (*E. coli*) at concentration of ~1 × 10⁹ colony-forming units (cfu)/mL was plated on an LB agar plate and incubated for 18 h at 37 °C to detect the existence of the growth inhibition halo around the samples.

*Bacterial growth kinetics test.* To evaluate the bacterial growth kinetics in the liquid medium of *E. coli*, samples were placed in the diluted bacterial suspension at the concentration of $10^5$–$10^6$ cfu/mL, which were subsequently cultured in a shaker incubator at 37 °C and 220 rpm. After predetermined times, the optical density, a measure of cell growth, was determined at 600 nm (OD600) using a microplate reader (Varioskan Flash, Thermo Scientific).

## Data availability
The datasets generated during and/or analyzed during this study are available from the corresponding author on request.

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

## Acknowledgements

We thank Mary Foss for antibacterial studies, Douglas Moffatt for strain sensor experimental setup and Mary Gallerneault for dipole antenna SEM analysis, TGA, and editing.

## Author contributions

B.D. and C.P. conceived the idea together and designed the experiments. H.W.d.H. devised and performed the simulations, as well as interpreted results. K.L.S. acquired the gelation rates, T.L. fabricated and measured the response of the strain sensors, and J. L. designed and setup the in-situ measurement of polymer network formation. The antenna measurements were performed by J.H. while the interpretation of the results were done by J.H. and R.E.A. J.T. performed the antimicrobial experiment. N.K and L.S.Y. printed, processed, and characterized printed cylinders. Material characterization was performed by B.D., C.P., K.L.S., and T.L. The manuscript was co-written by B.D., C.P., K.L.S., T.L., J. L., J.H., H.W.d.H., J.T., and P.R.L.M.

## Competing interests

The authors declare no competing interests.
