## [Peer Review File · Nature Communications]

REVIEWER COMMENTS

Reviewer #1 (Remarks to the Author):

The manuscript of Deore et al. describes a clever method to generate silver-containing polymeric 3D object in a single-step procedure. While 3D printing is about to revolutionize manufacturing, there are some challenges remaining with regard to the materials that are amenable to printing and more importantly their combinations. The present manuscript describes a new method based controlling the diffusion of one of the components (the silver precursor AgND) by phase-separation and the onset of network formation. First, the authors analyze the intricate balance between these factors by varying concentration of the AgND as well as the chemical structure of the crosslinker. Based on their analysis, they are able to control the diffusion of the AgND and thus subsequent formation of silver particles (and later on silver phases). Finally, they take advantage of the unique electronic as well antimicrobial properties of silver to demonstrate the use of their technique to print strain sensors, 5G antennae, and bacterial growth-inhibiting surfaces.

The approach outlined in this work is original and the work has been carried out with great care. I could not identify weaknesses and thus recommend acceptance of the manuscript in Nature Communications.

Reviewer #2 (Remarks to the Author):

Deore et al. report on a simple and effective method to prepare UV-curable thermosets with fairly well-controlled silver gradients. Notably, under certain conditions the authors show that sufficiently high concentrations of silver can be formed at the surface of the material such as to generate a conductive material. Finally, the authors demonstrate the board utility of their material through 3D printing of complex objects (Figure 1), as strain sensors (Figure 5), for wave antennas (Figure 6) and to have general antibacterial properties (SI).

We felt the references were generally thorough. Although we would like to draw authors attention to the follow paper <https://doi.org/10.1016/j.jallcom.2019.03.026> published last year in Journal of Alloys and Compounds. In this work a ZrOCr/resin suspension was vat-polymerized/printed and then sintered to arrive at parts with inner porosity and a solid outer core. This general morphology is strikingly similar to that of the AgND structures in this work. Is this a general phenomenon, i.e. phase separation induced migration of particles/metal nano-particles towards the surface?

General Questions/Comments

- 1) There was no discussion of Au or Cu. Can this methodology be extended to group 11 metals?
- 2) The cross-sectional SEM images in Supplementary Figure 2 were very illuminating and in our opinion, more informative than cylinder graphic in Fig 1B. We therefore strongly recommend moving one of the supplementary SEM images to Figure 2 to better explain the Figure 2A SEM images.
- 3) In the section entitled "Crosslinkers drive the spatial distribution of silver" the authors should describe the nominal wt% of Ag present in the resists within the discussion of resin composition. This information is provided in the experimental section, but just a quick note that says all resists contain 25wt% silver would help the reader.
- 4) Similar comment to the one above, the caption in Figure 2 should contain similar information describing the amount of silver and as well as a note that all of the cylinders were sintered prior to measurement.
- 3) In Figure 2D, how was resistance measured? This information should be included in the

Experimental section

3a) As it relates to resistance, do the samples need to be sintered? A further discussion on the role of sintering would be welcomed.

4) The resistance/conductivity of these materials should be compared to other examples in the literature, i.e. there should be a discussion somewhere in the body of this manuscript. From a quick google search, DOI:10.1007/s11433-016-0447-6 this review on metallized structures focuses heavily on achieved high conductivities.

Further, the authors note that their method is superior to “two-step coating methodologies” but this claim needs to be substantiated. The authors suggest a comparison in Supplementary Figure 4, but there is no information on how the electroless deposition step was performed.

5) Briefly in the experimental section, the authors note that although they utilized resins with ~25% Ag, it is possible make a resin with 50% Ag. What effect does Ag loading have on resistivity?

6) Figure 1C – please provide a scale bar for the lattices.

7) What was the light intensity (in mW/cm²) used to print the cylinder/material (for the experiments shown in Figure 2) and the also what was the light intensity of the 405 nm laser used to preform the gelation experiments shown in Figure 3? Also what was the dose (intensity*time) required to generate the printed objects in this work, specifically the cylinders utilized in Figure 2?

8) How does light intensity affect gelation? How does intensity affect phase separation.

8) If the intensities in question 7 were different, please explain how the gelation kinetics in Figure 3 that were then applied to the simulations in figure 4, were valid to explain the phenomenon observed in figure 2. The arguments for the observed phase separation revolve around rates of gelation and the intensity of the light source is well-known to influence the rate of gelation for photopolymerization. We very much understand that it is difficult to make kinetic measurements of photopolymers under relevant vat-polymerization conditions, especially if two different light sources are used. But this needs to be acknowledged and discussed.

9) Figure 3C contains gelation data plotted against surface Ag wt%. Was all of the data in figure 3C obtained from experiments using the 405 nm laser? The authors state that the data in Figure 3B was obtained from resins sans Ag? Why not utilized Ag-loaded resins to generate the data shown in 2B.

10) There is a variable that the authors do not discuss that very well could be playing a large role in the observed kinetics and that is the concentration of inhibitor. All acrylate monomers contain some amount of inhibitor, usually MEHQ (4-methoxyquinone). The authors do not list where they obtained the chemicals used in this work (please provide commercial sources). Assuming the monomers were obtained from sigma Aldrich, Ethylhexyl Acrylate contains 1100 ppm MEHQ (<https://www.sigmaaldrich.com/catalog/product/aldrich/290815?lang=en&ion=US>), whereas PEGDA 250 contains 100 ppm MEHQ. The ratio of inhibitor to initiator will drastically effect gelation and polymerization rates. Since the resins have varying amounts of PEGDA to Ethylhexyl Acrylate, the authors should account inhibitor concentration.

11) the initiator in this work, TPO-L, should be listed in the experimental section. Currently it can only be found in the SI, the experimental section simply says “initiator”.

12) The paper described the length of the simulated monomers as ‘L = 3, 6, 9’. Figure 4 says N = 3,6,9.

13) It is very difficult to tell what is happening in Figure 4b,c. Clearly there is a molecular (green balls) trapped in a network, but beyond that what is Figure 4b,c trying to communicate?

14) On page 17, the authors write: "This density is high enough to approximate the pure polymer melt of the experiments, but is also low enough to allow for the movement of individual crosslinker molecules on simulation time scales." What is meant by "high enough and low enough", please cite relevant sources.

15) in the section entitled "3D PIPS smart objects" – the authors describe a truss structure, is this the structure shown in Figure 1C? A image of the truss structure would help clarify the SEM images in Figure 5.

16) The authors also write, "formulations yield thermosets with low stiffness". Please clarify what is meant by "low stiffness". Please provide compression data or other evidence.

17) In general, using stress-sensors that operate on electrical conductivity of nanoparticles (graphene, metals, etc.) impeded in a polymer matrix is well known and widely utilized strategy. This section could benefit from further discussion, specifically in the context of sensitivity of their devices in comparison to other published work. A quick google search turns up numerous reviews, including doi: 10.1186/s11671-019-3084-x.

18) the image in Figure 6B is too small to read. Please provide a scale bar for the picture in 6A.

19). In the experimental section, the authors write: "These resin formulations are comprised of 8.0 wt % of silver metal or 9.5 wt % silver metal post-sintering assuming neodecanoate and 2-ethyl-oxazoline become volatile during the process." This could be confirmed via thermal gravimetric analysis or simply weighing the sample after sintering.

Reviewer #3 (Remarks to the Author):

Deore et al report a method to 3D print multi-material silver composites that utilize phase separation to create different spatial distributions of silver to afford different morphologies. The fabrication of multi-material constructs via 3D printing represents a significant challenge to the field. In particular, there are very few strategies for multi-material vat photopolymerization that have been developed, which the authors appropriately cited. In this manuscript, the authors propose a method for making silver composites with different morphologies based on polymerization-induced phase separation. The authors propose that the silver migration through the matrix material is dependent upon the kinetics of gelation, polymer network density, and material diffusivity.

While the concept is interesting, this manuscript does not meet my expectations for advancements in the field reported in this journal. Moreover, the authors left many unanswered questions that are important to supporting the claims made in this manuscript.

1) Polymerization induced phase separation (PIPS) typically involves phase separation as a result of polymer chain growth, and this does not appear to be the case in the work. In the case of this manuscript, the samples are sintered at 210 C to convert AgND precursors into metallic silver. The

authors did not provide clear evidence that AgND is phase separated prior to sintering. So how do the authors know that their observations are not due to different kinetics of polymer degradation? The different compositions of the resin could afford different rates of degradation, which could also affect the phase separation that is observed. Along the same lines, what was the mass loss after sintering?

2) Why are there not nano- and micro-islands of silver metal within the polymer network? It is unclear why silver would migrate to the air interface.

3) Pg 5, ln 22: The authors should define exactly what these percentages represent.

4) The authors mention a "probe molecule" on page 9 but never stated the identity of the compound.

5) Pg 9: The authors discuss how shorter crosslinkers afford higher density of crosslinking, which leads to smaller pore size. The authors should clarify whether they are referring to pore size or mesh size.

6) The explanation of how the strain sensor works on page 11 was unclear, particularly in reference to Figure 5b.

7) Some control experiments for the strain sensor, antennae, and antimicrobial surface would have improved the impact of these experiments.

8) Figure 1 is unclear to me. It is difficult to tell what we are supposed to see, particularly in b. The images in c should have a scale bar or reference.

9) Figure 6 should use a larger font size. Also the image in 6a is difficult to see. Was this 3D printed?

10) In Supplementary Figure 3 needs information about the scale bar and a sample of SEM image and EDS layer image cannot read. The SEM image of 1.0 wt % Ag sample is missing, in Supplementary Figure 8. It is hard to see the blue fluorescence regions of EDS images, f, and g in Supplementary Figure 10.

REVIEWER COMMENTS

Reviewer #1 (Remarks to the Author):

The manuscript of Deore et al. describes a clever method to generate silver-containing polymeric 3D object in a single-step procedure. While 3D printing is about to revolutionize manufacturing, there are some challenges remaining with regard to the materials that are amenable to printing and more importantly their combinations. The present manuscript describes a new method based controlling the diffusion of one of the components (the silver precursor AgND) by phase-separation and the onset of network formation. First, the authors analyze the intricate balance between these factors by varying concentration of the AgND as well as the chemical structure of the crosslinker. Based on their analysis, they are able to control the diffusion of the AgND and thus subsequent formation of silver particles (and later on silver phases). Finally, they take advantage of the unique electronic as well antimicrobial properties of silver to demonstrate the use of their technique to print strain sensors, 5G antennae, and bacterial growth-inhibiting surfaces.

The approach outlined in this work is original and the work has been carried out with great care. I could not identify weaknesses and thus recommend acceptance of the manuscript in Nature Communications.

Reviewer #2 (Remarks to the Author):

Deore et al. report on a simple and effective method to prepare UV-curable thermosets with fairly well-controlled silver gradients. Notably, under certain conditions the authors show that sufficiently high concentrations of silver can be formed at the surface of the material such as to generate a conductive material. Finally, the authors demonstrate the board utility of their material through 3D printing of complex objects (Figure 1), as strain sensors (Figure 5), for wave antennas (Figure 6) and to have general antibacterial properties (SI).

We felt the references were generally thorough. Although we would like to draw authors attention to the follow paper <https://doi.org/10.1016/j.jallcom.2019.03.026> published last year in Journal of Alloys and Compounds. In this work a ZrOCr/resin suspension was vat-polymerized/printed and then sintered to arrive at parts with inner porosity and a solid outer core. This general morphology is strikingly similar to that of the AgND structures in this work. Is this a general phenomenon, i.e. phase separation induced migration of particles/metal nano-particles towards the surface?

We thank the referee for bringing this example to our attention. Upon assessing the paper, we agree that it is possible that phase separation of the zirconium precursor takes place as it does have some of the hallmark features, such as an enrichment of zirconium with respect to carbon at the surface versus the core. However, because the authors do not characterize the print prior to

pyrolyzing the sample, it is difficult to confirm whether this feature forms as a result of pyrolysis or due to phase separating precursors during printing.

General Questions/Comments

1) There was no discussion of Au or Cu. Can this methodology be extended to group 11 metals?

In principle it is possible to apply these concepts to Au or Cu molecular precursors and nanoparticles. The phase separation of smaller precursors will be less impeded than with larger nanoparticles as they will not become trapped by the polymer network to the same extent as the nanoparticles. We did some preliminary work using copper formate precursors which demonstrated evidence of phase separation however as copper is highly sensitive to oxidation, we were not able to produce conductive samples. Based on our previous work in 2D printed electronics with copper, thicker films of copper are often required to mitigate the effects of oxides on the electrical properties of copper films. Therefore, further work would be required to optimize the conditions to get a thicker film of copper precursor at the surface of the 3D printed objects or to introduce a reducing environment during sintering.

2) The cross-sectional SEM images in Supplementary Figure 2 were very illuminating and in our opinion, more informative than cylinder graphic in Fig 1B. We therefore strongly recommend moving one of the supplementary SEM images to Figure 2 to better explain the Figure 2A SEM images.

As per reviewer's suggestion, we moved the SEM images of Supplementary Figure 2 into the manuscript (Figure 2a). The figure caption is revised as written below (see page number 34). We also updated the SEM images in Figure 1B to more clearly show the difference between coating, gradient and composite. The following additions were made to the figure caption and main text:

New text in the Figure caption on page 34:

Figure 2. Silver phase separation as a function of crosslinking density. *Cylinders 1.5 mm in diameter and 2 cm in length were printed using resins containing 25 wt % AgND. a) SEM cross-sectional images of cylinders printed using resin mixtures (i, ii) 25 wt % DA-170 and (iii, iv) 25 wt % DA-250 and taken prior to thermal sintering, but treated with 5 minutes of UV curing to convert some of the silver salt to silver metal. These results confirm that the silver complex diffuses to the surface during printing and the partitioning of material phases does not happen post-printing. The cylinders once sintered at 210 °C contain 9.5 wt % Ag. b) Cross-sectional SEM images focused approximately < 5 μm from the edge of the cylinder. Cylinders were printed using 25, 50 and 99 wt % for DA-170, DA-250, DA-575 and DA-700 crosslinkers. c) Weight fraction of silver as a function of depth with respect to the surface as measured by performing EDS on the surface of the object (data point at 1 μm) and on the cross-sections of the cylinders for 25, 50 and 99 wt % crosslinker of each DA type. d) Surface silver of sintered cylinders as a function of wt % crosslinker as measured by EDS analysis on the surface of the object. The estimated interaction volume of the EDS beam for these measurements performed on the top surface of the object is ~2 μm. e) Resistance as a function of wt % crosslinker for post-sintered cylinders.*

New text in the manuscript on page 6: *The morphologies formed by the various resins, both pre- and post-sintering, can be seen in the SEM images taken at the edge of the cross-sections of the cylinders (Figure 2a and b and Supplementary Figure 3). The coating of the pre-sintered 25 wt % DA-250 and DA-170 cylinders confirm that PIPS occurs during printing.*

New text in the Supporting Information on page 9: *To verify that silver migrates during the printing process, a formulation known to generate a coating morphology was 3D printed into a cylinder and UV sintered immediately after printing. UV sintering has the effect of converting the silver salt to silver nanoparticles and, thus, by applying UV light to the cylinder, the silver is effectively trapped in space. As shown in the SEM images of Figure 2a in the manuscript, the bright areas at the surface of the cylinders indicate that silver is present at the surface. In a similar experiment, cylinders were printed and left over several days, allowing 2-ethyl-2-oxazoline to evaporate and allowing SEM imaging under reduced pressures. As shown in Supplementary Figure 3, SEM and EDS analysis confirm that the silver is concentrated at the surface of the cylinders.*

3) In the section entitled "Crosslinkers drive the spatial distribution of silver" the authors should describe the nominal wt% of Ag present in the resists within the discussion of resin composition. This information is provided in the experimental section, but just a quick note that says all resists contain 25wt% silver would help the reader.

In the section entitled "Crosslinkers drive the spatial distribution of silver," we describe the nominal wt % of Ag present in the resin on page 6 in the revised text.

***New text on page 6:** 3D PIPS was first demonstrated by printing cylinders 1.5 mm in diameter and 2 cm in length and sintered at 210 °C to convert AgND into metallic silver. These resins contained 25 wt % AgND yielding objects with 9.5 wt % silver post-sintering (see Supplementary Figure 2 and Supplementary Table 3). The morphologies formed by the various resins, both pre- and post-sintering, can be seen in the SEM images taken at the edge of the cross-sections of the cylinders (Figure 2a and b and Supplementary Figure 3).*

4) Similar comment to the one above, the caption in Figure 2 should contain similar information describing the amount of silver and as well as a note that all of the cylinders were sintered prior to measurement.

The nominal amount of wt % Ag present in the resin and sintering details are added in the Figure 2 caption. See new caption for Figure 2.

3) In Figure 2D, how was resistance measured? This information should be included in the Experimental section.

Detailed information of the resistance measurements are included in the experimental section on page 17.

***New text on page 17:** The resistance (R) values were taken by measuring on the walls of the cylinders with the probes separated by 1 cm. Lead contact resistance was found to be negligible, and therefore, a two probe measurement method was used. Average resistances were obtained by measuring 10 different cylinders. Sheet resistances were calculated based on the circumference of the cylinder (W) and the length (L) between electrical probes (1 cm), where the sheet resistance (R_s) is $R_s = R \times W / L$.*

3a) As it relates to resistance, do the samples need to be sintered? A further discussion on the role of sintering would be welcomed.

Yes, the samples do need to be sintered in order to convert the silver salt to a metallic film. The discussion on the role of sintering is elaborated on in the manuscript on page 4 and 5.

***New text on page 4 and 5:** The silver precursor, a mixture of silver neodecanoate with 2-ethyl-2-oxazoline (AgND), is ideal for this application as its molecular nature ensures higher diffusivity*

*than larger functional materials such as nano- or micro-particles. Furthermore, because AgND does not scatter light as particles do, resins containing high concentrations of the complex can be printed. As has previously been shown with screen printable inks derived from this salt, the precursor will decompose into volatile products and conductive metallic silver traces with volume resistivity values as low as $9 \mu\Omega\cdot\text{cm}$ through a simple post printing sintering step using temperatures greater than 150°C .*³⁸

4) The resistance/conductivity of these materials should be compared to other examples in the literature, i.e. there should be a discussion somewhere in the body of this manuscript. From a google search, DOI:10.1007/s11433-016-0447-6 this review on metallized structures focuses heavily on achieved high conductivities.

During the preparation of the manuscript, we carefully considered the best way to express the electrical properties of these structures. The ideal measure is volume resistivity as it allows for a direct comparison with other examples in literature. Previous examples of 3D printed conductive composites (based on conjugated polymers, graphene and silver) exist and they have stated volume resistivity values based on the dimensions of the 3D printed structure [Erika Fantino et al., Adv. Mater. 2016, 28, 3712–3717 and Xiaolu Wang et al., ACS Appl. Mater. Interfaces 2019, 11, 21668–21674]. In our work, volume resistivity calculated using a similar approach (i.e. based on the dimensions of the printed objects) would not provide a meaningful value due to the morphology of our samples, that is, one where the conductivity originates from the surface coating of the printed object. If we were to calculate the volume resistivity of our structures using the dimensions of the printed object in the same fashion as the above references, our values would be dependent on geometry of the printed object and would not reflect the electrical conductivity of the silver coating.

The volume resistivity of the silver layer could be calculated using the thickness of the silver coating, however it would only be an estimation as the graded compositions of the silver makes it difficult to determine the thickness of the conductive layer. We feel the best approach would be to provide values of sheet resistivity and compare the value to literature examples of 2D printed silver. The text of the manuscript has been modified to elaborate on our calculation of sheet resistance and provide literature comparisons.

(from question 3) New text in experimental section on page 17: *The resistance (R) values were taken by measuring on the walls of the cylinders with the probes separated by 1 cm. Lead contact resistance was found to be negligible, and therefore, a two probe measurement method was used. Average resistances were obtained by measuring 10 different cylinders. Sheet resistances were calculated based on the circumference of the cylinder (W) and the length (L) between electrical probes (1 cm), where the sheet resistance (R_s) is $R_s = R \times W / L$.*

New text on page 8:

The calculated sheet resistance of $340 \text{ m}\Omega/\square$ is commensurate with values reported by Kell et al. in which screen printed traces using the same silver precursor have sheet resistance values of $\sim 200 \text{ m}\Omega/\square$. The values are three orders of magnitude lower than recently developed 3D

printable conjugated polymers with values of $\sim 6.6 \times 10^5 \text{ m}\Omega/\square$.³⁹

Further, the authors note that their method is superior to “two-step coating methodologies” but this claim needs to be substantiated. The authors suggest a comparison in Supplementary Figure 4, but there is no information on how the electroless deposition step was performed.

We have added details that describe how the method was optimized for electroless deposition.

New text on page 16:

Preparation of Dipole Antenna by Electroless Silver Plating: *The mixed DA-resins were prepared by mixing different volumes of two separately prepared DA-resins such as 75 mL of 50 wt % DA-575 and 25 mL of 35 wt % DA-250. The dipole antennas were printed using the silver-free resin and subsequently seeded with silver particles to yield an adherent and uniform silver coatings on the antenna substrate. For a seeding pre-treatment procedure, antennas were dip coated with 100X dilute solution of commercial nanoparticle ink (SunTronic™ NANOSILVER) in toluene. The thin layer of seeds did not result in any measurable conductivity; however, this step was crucial to obtain uniform and adherent silver coating through electroless plating. Further, electroless plating procedure was optimized by varying the rate of addition and concentration of the silver plating bath solutions in order to obtain coatings with low surface roughness.⁴⁸ The electroless silver plating bath was composed of solution A (0.2 g of glucose, 0.02 g of tartaric acid and 0.5 mL of ethanol in 20 mL of deionized water) and solution B, a $\text{Ag}(\text{NH}_3)_2^+$ solution (0.2 g of AgNO_3 , 0.075 g of NaOH and 0.5 mL of ammonia in 20 mL of deionized water). Over a period of 60 mins, under constant stirring, these two solutions were combined by dropwise addition of solution B into solution A in a bath containing the dipole antenna at room temperature. The antennas were then collected, rinsed with water and dried at 140 °C for 5 minutes. The resistance and the adhesion of the silver coatings of the samples prepared with the optimized electroless plating conditions were compared to those obtained through phase separation (Supplementary Figure 5). The adhesion of the silver coatings was determined by applying tape to the sample, pressing firmly and removing the tape.*

New Supplementary Figure 5 and text in SI on page 11:

A comparison of antennas made using the phase separation and electroless plating methods is found in Supplementary Figure 5. Optical profilometry measurements (CT-100 by Cyber Technologies) were performed on the surface of each sample to determine relative roughness of each method. The root-mean-square surface roughness (Rq) values of 0.24 and 4.53 μm were found for the antennas prepared by phase separation and electroless plating of silver, respectively. Photographs of the antennas before and after an adhesion tape test show that the antennas prepared by phase separating silver forms a silver coating with significantly improved adhesion than that of the silver coating generated by electroless plating.

Root-mean-square surface roughness (Rq)

0.24 μm

4.53 μm

Supplementary Figure 5. SEM images of the surface of dipole antenna prepared using (a, b) phase-separated and (c, d) electroless plating. The images show the higher surface roughness of the electroless-plated sample compared to the phase separated samples. Roughness values found using optical profilometry show the phase separated samples are one order of magnitude lower than that of electroless deposited samples. Optical microscope images of the samples after a tape adhesion test showing the improved adhesion of the silver coating on samples prepared using phase separation (e) compared to electroless plating (f).

5) Briefly in the experimental section, the authors note that although they utilized resins with ~25% Ag, it is possible to make a resin with 50% Ag. What effect does Ag loading have on resistivity?

It is possible to print structures with 50% AgND however the electrical and mechanical properties of the structures begin to deteriorate at this concentration. The table below, showing the resistance of cylinders as a function of the wt % of Ag for resins made with 35% DA-250 (and included in the Supplementary Information as Table 3 on page 7) demonstrates that there is an optimal range of AgND loading for achieving low resistivity. A minimum concentration of silver is required to produce conductive silver coatings with the electrical conduction achieved at a threshold concentration of ~19 wt % AgND. The resistance of the surface of the cylinders did not change significantly when the concentration of AgND in the resin was between 25 and 38 wt %, while AgND concentrations greater than 38 wt % reduced the printability of the resin and resulted in brittle objects and cracked silver coatings (Supplementary Figure 1). We modified the manuscript to make mention of this Table in the main text.

New table in the Supporting Information on page 6:

Supplementary Table 1. Resistance as a function of different wt % of silver precursor (AgND, silver neodecanoate and 2-ethyl-2-oxazoline) in 35 wt % crosslinker DA-250 formulations and their sintered samples.

wt % AgND**** precursor in formulation*	wt % Ag in formulation**	wt % Ag in sintered sample***	Resistance per cm (Ω)	Sheet resistance Ω/\square
6.25	1.97	2.38	Not conducting	-
12.50	3.94	4.75	Not conducting	-
18.75	5.91	7.16	20.0 ± 5	6.3
25.00	7.88	9.50	1.3 ± 0.1	0.4
30.11	9.49	11.88	0.9 ± 0.1	0.3
37.50	11.82	14.25	3.6 ± 0.7	1.1
50.00	15.76	19.00	Poor electrical and mechanical properties	-

*pre-sintering silver neodecanoate content (column 1):

$$\text{wt \% AgND} = \text{wt AgND} / [\text{wt AgND} + \text{wt resin}] \times 100\%$$

**pre-sintering silver content (column 2):

$$\text{wt \% Ag} = \text{wt Ag} / [\text{wt AgND} + \text{wt resin}] \times 100\%$$

***post-sintering silver content (column 3):

$$\text{wt \% Ag} = \text{wt Ag} / [\text{wt Ag} + \text{wt resin}] \times 100\%$$

****AgND defined as silver neodecanoate with 2-ethyl-2-oxazoline in weight ratios 1:0.22.

New text in the manuscript on page 5:

A concentration of 25 wt % silver precursor (AgND) was used throughout the study based on results showing optimal electrical conductance at this concentration (Supplementary Table 3). A threshold concentration of ~19 wt % AgND was required for electrical conduction using the 35 wt % DA-250 formulation. The resistance of the surface of the cylinders did not change significantly when the concentration of AgND in the resin was between 25 and 38 wt %, while AgND concentrations greater than 38 wt % reduced the printability of the resin and resulted in brittle objects and less uniform silver coatings (Supplementary Figure 1).

New Figure in the Supporting Information on page 7:

Supplementary Figure 1. Optical microscope images of cylinder printed using 25, 30 and 38 wt % AgND and using resins containing 35 wt % crosslinker DA-250.

6) Figure 1C – please provide a scale bar for the lattices.

To provide a scale for the lattices in Figure 1C, we added the dimensions of the object to the caption.

New text on Page 32:

Figure 1. c) Photographs of PIPS induced 3D printed and sintered Ag coated lattice objects with dimensions of 40 x 40 x 40 mm.

7) What was the light intensity (in mW/cm²) used to print the cylinder/material (for the experiments shown in Figure 2) and the also what was the light intensity of the 405 nm laser used to preform the gelation experiments shown in Figure 3? Also what was the dose (intensity*time) required to generate the printed objects in this work, specifically the cylinders utilized in Figure 2?

Based on the spot size provided by the manufacturer and the measured laser power, we calculated the light intensities used in the printer and for the gelation experiment. We also include the dose based on the print speed. All data in the manuscript was collected using a laser power of 760 W/cm²; however, as the laser power on the printer is tunable, we have included data at two additional light intensities to allow further discussion (question 8). More specifically, we printed cylinders using 350 and 560 W/cm² laser intensities. When printing using light intensities below 350 W/cm², the laser intensity was insufficient to produce 3D printed structures with the resin formulations used in the manuscript. These values are all included in a new section dedicated to the effect of light intensity in the Supplementary Information (see answer to question 8 and 9). New text was also added to the experimental section as shown in the underlined parts below.

New section in the Supplementary Information on page 21 and 22:

Light intensities

Different light intensities for 3D printing were explored in order to establish optimum printing conditions. The light intensities tested were 350, 560 and 760 W/cm² as shown in Supplementary Table 4. The resistances of the cylinders printed under these light intensities are shown in Supplementary Figure 14 and show no significant differences. These results suggest that light intensity does not affect the phase separation of silver significantly within the range of 350 to 760 W/cm².

The gelation experiments that were used to acquire delay times employed a light intensity of 566 W/cm² and falls within the range of light intensities used for printing. As the resistance did not vary significantly within this range, the gelation behavior of the resins is expected to be the same for printing and gelation experiments, thus, validating our use of delay times as a relative measure of gelation rates for 3D printed objects. The gelation experiments were done under constant illumination/no moving laser rate so dose couldn't be calculated.

Supplementary Table 4. Light intensity and estimated light doses of the laser used for printing and gelation experiments.

	Printing (laser setting 58)	Printing (laser setting 66)	Printing (laser setting 75)	Gelation experiment
Spot radius (μm)	35	35	35	0.75
Spot radius (cm)	0.0035	0.0035	0.0035	0.000075
Spot area (cm ²)	3.85E-05	3.85E-05	3.85E-05	1.77E-08
Laser power (mW)	13.5	21.7	29.2	0.01
Light intensity (W/cm ²)	350	560	760	566
Linear printing rate (cm/s)	8.5	8.5	8.5	N/A
Light dose (mW•s/cm ²) ^[1]	360	580	780	N/A

[1] Zakeri, S., Vippola, M. & Levänen, E. A comprehensive review of the photopolymerization of ceramic resins used in stereolithography. *Addit. Manuf.* 35, 101177 (2020).

Supplementary Figure 14. Resistance of cylinders printed using different light intensities for DA-250 resins.

New text in the Experimental Section on page 17-18:

Printing, Sintering and Characterization of 3D Objects: SLA Printing of 3D Objects: 3D objects using functional material were printed using Peopoly Moai Laser SLA 3D Printer (Technical Specifications: Build Volume: 130 x 130 x 180 mm, Laser spot size: 70 microns, Laser wave length: 405 nm, Laser power: 150 mW (measured power output 29.2 mW), Machine size: 330 x 340 x 660 mm, Layer Height: 10 – 200 microns. Objects were printed using a fluorinated ethylene propylene liner-coated vat with laser power rating set to 75 and all other settings set to the default. All cylinders were printed using light with an intensity of 760 W/cm² other than optimization experiments where 350 and 560 W/cm² were used to explore the role of light intensity on the phase separation of silver as described in the Supplementary Information (Supplementary Table 4 and Supplementary Figure 14).

Optical Studies of Resin Polymerization Kinetics: The photopolymerization was studied using an experimental method employing an optical microscope to observe the changes in refractive index which occurs as the resin polymerizes under illumination by a 405 nm laser as shown in Supplementary Figure 16. Each resin was prepared in 2 – 10 mL quantities (see Supplementary Table 2 for formulations) and loaded into a capillary micro glass slide (0.10 x 2.0 mm, 0.10 mm wall thickness; Electron Microscopy Sciences). The filled capillary micro glass slide was placed under the optical microscope (Nikon ME600) fitted with a 405 nm laser and optical filters (405 nm laser line and 405 nm Raman edge filters) to prevent saturation of the camera with unwanted laser light. Laser power (10 μW with 1-2 μm diameter spot size and ~566 W/cm² intensity) was set low enough to reveal polymerization in a 10 – 60 s timescale.

8) How does light intensity affect gelation? How does intensity affect phase separation?

Rate of polymerization, R_p , varies with light intensity, I , as $R_p \sim \sqrt{I}$. [Odian, Principle of Polymerization 4th Edition] The rate of gelation (R_G) is directly linked to the rate of polymerization and, thus, the rate of gelation will vary as $R_G \sim \sqrt{I}$. [Szczepanski, Polymer 2012] Therefore, higher light intensities will increase gelation rates by increasing polymerization rates. As mentioned above, we printed samples at two different light intensities for three different crosslinker concentrations using the DA-250 crosslinker. For this range of intensities (350 to 760 W/cm²), the effect of light intensity on the resistivity is not significant, and therefore, we assume that the light intensity does not significantly affect the gelation behavior and phase separation within this range of light intensities. This is discussed in the new section of the Supplementary Information on light intensity.

8) If the intensities in question 7 were different, please explain how the gelation kinetics in Figure 3 that were then applied to the simulations in figure 4, were valid to explain the phenomenon observed in figure 2. The arguments for the observed phase separation revolve around rates of gelation and the intensity of the light source is well-known to influence the rate of gelation for photopolymerization. We very much understand that it is difficult to make kinetic measurements of photopolymers under relevant vat-polymerization conditions, especially if two different light sources are used. But this needs to be acknowledged and discussed.

As we have shown in answering question 7, the amount of surface silver can be validly compared with delay times since the light intensities used are similar during printing and the gelation experiment. Furthermore, the delay times that result from the gelation experiments are relative. If any differences in light intensity caused changes in gelation times, it would not cause changes to the relative delay times between resins, and therefore, would not impact any of the conclusions drawn from the data.

The simulations are used to help understand why the four series in Figure 3 do not collapse into a universal curve. The simulation are specific in their goal to compare the diffusivity of the silver salt (or probe molecule) in two scenarios: in the resin and in a fully formed polymer network. The simulations do not attempt to model the diffusion of the probe molecule in a dynamic state. Thus, the simulations do not use rates of gelation as inputs, but rather only the initial state (monomer) and final state (polymer network).

The following text was included to provide clarity on this point.

New text in manuscript on page 9-10:

To explore this idea, coarse-grained Langevin dynamics simulations of a simplified system were performed (see Methods for details). The simulations tracked the displacement of a probe molecule, representing silver neodecanoate, in systems containing 100 wt % crosslinkers. The simulations monitored the diffusivity of the probe molecule in unreacted resin and in a fully formed polymer network only, and therefore, the simulations did not require a consideration of kinetic effects of the polymerization reaction.

9) Figure 3C contains gelation data plotted against surface Ag wt%. Was all of the data in Figure 3C obtained from experiments using the 405 nm laser? The authors state that the data in Figure 3B was obtained from resins sans Ag? Why not utilized Ag-loaded resins to generate the data shown in 2B.

To provide clarity, we have removed “silver migration” from the title of Figure 3 and added the following to the caption in Figure 3c: “The delay times are from the same gelation experiments exposed to a 405 nm laser in part a) and b) for each formulation.”

The gelation experiments were completed without AgND as the presence of the silver salt made it more difficult to determine the interface between the polymer island and the remaining resin in the capillary tube. We were able to measure a few samples of DA-575 with Ag and observed that the trend in delay time with respect to wt % crosslinker was the same, albeit ~ 1.0 to 2.0 seconds slower. This is more clearly detailed in the manuscript on page 35, in the Supplementary Information on page 12, and in the Experimental Section on page 18.

New Caption on page 35:

Figure 3. Dynamical study of photoresins by optical microscopy. a) Constructed images obtained from videos of islands of polymer formed when resin/crosslinker inside a capillary is exposed to a focused 405 nm laser spot. A slice through the center of the island reveals its two opposite edges and is plotted as function of time. Two examples i) and ii) are shown for 15 wt % DA-170 and 99 wt % DA-700. b) Delay time, t_d , extracted from a fit to Equation 1 (See Experimental Section) as a function of wt % crosslinker. c) Weight % of Ag extracted from SEM/EDS analysis at the surface of the cylinder as a function of t_d for various resin formulations. The delay times are from the same gelation experiments exposed to a 405 nm laser in part a) and b) for each formulation.

New text in caption of Supplementary Figure 6 on page 12:

We note that experiments performed using photoresins with AgND show similar trends in t_d as a function of wt % of crosslinker compared to those performed without AgND.

New text in Experimental Section on page 18:

Each resin was prepared in 2 – 10 mL quantities (see Supplementary Table 2 for formulations) and loaded into a capillary micro glass slide (0.10 x 2.0 mm, 0.10 mm wall thickness; Electron Microscopy Sciences). The resins were prepared without AgND as its presence made it more difficult to clearly determine the interface between the polymer island and remaining resin in the capillary tube, but a similar gelation time trend was found between resins with and without AgND for the DA-575 crosslinker (Supporting Information Figure 6).

10) There is a variable that the authors do not discuss that very well could be playing a large role in the observed kinetics and that is the concentration of inhibitor. All acrylate monomers contain some amount of inhibitor, usually MEHQ (4-methoxyquinone). The authors do not list where they obtained the chemicals used in this work (please provide commercial sources). Assuming the monomers were obtained from sigma Aldrich, Ethylhexyl Acrylate contains 1100 ppm MEHQ

(<https://www.sigmaaldrich.com/catalog/product/aldrich/290815?lang=en&ion=US>), whereas PEGDA 250 contains 100 ppm MEHQ. The ratio of inhibitor to initiator will drastically effect gelation and polymerization rates. Since the resins have varying amounts of PEGDA to Ethylhexyl Acrylate, the authors should account inhibitor concentration.

The inhibitors were left in the monomers and crosslinkers. The concentration of inhibitor in the crosslinkers ranged from 0 to 0.006 wt % while the monomer, EHA, contained 0.004 to 0.11 wt % of inhibitor. We chose to keep the inhibitor in the resins in order to ensure greater consistency from print to print as we concluded the inhibitor does not alter the relative trends in t_d and, as a result, the conclusions drawn in the manuscript.

When the concentration of inhibitor is high relative to the initiator, the inhibitors will retard polymerization, affecting rates of polymerization which in turn will affect the delay times. In our experiments, the initiator is at a minimum 6 times that of the inhibitor. An indication that the inhibitor does not have a pronounced affect is Figure 3b. More specifically, the 100% DA-170 does not contain any inhibitor while the other series contain inhibitor. Therefore, we would expect this series to have a shorter delay time relative to other series if the inhibitor had a significant effect on gelation rates. When compared with 100% DA-250, DA-575 and DA-700 which have inhibitor, the 100% DA-170 has the longest delay time. We believe the higher concentration of initiator with respect to inhibitor resulted in the inhibitor playing an insignificant role in the delay times.

In addition, in our experiments, we account for the variation that the inhibitor could introduce by measuring the effective delay times, i.e. includes all factors that can affect gelation rates. Thus, any effect that the inhibitor has on the delay times will also have on the phase separation of the silver salt. Nevertheless, we agree clarification on role of the inhibitor should be made. We have included the following text to the experimentation section

New text in the Supplementary Information on page 4:

*The monomer and crosslinkers were purchased from Sigma Aldrich (Oakville, ON, Canada). The concentration of inhibitor in the crosslinkers ranged from 0 to 0.006 wt % while the monomer, EHA, contained 0.004 to 0.11 wt % inhibitor. The inhibitor was left in the resins in order to ensure greater consistency from print to print. A molar concentration of photoinitiator of at least 6 times that of the inhibitor was used to ensure the inhibitor did not have a significant effect on polymerization rates. The photoinitiator, ethyl (2,4,6-trimethylbenzoyl) phenylphosphinate (TPO-L), and 2-ethyl-2-oxazoline was purchased from Sigma Aldrich. The silver neodecanoate was prepared as described in the following reference: Titkov, A. I., Logutenko, O. A., Bulina, N. V., Yukhin, Y. M. & Lyakhov, N. Z. Synthesis of nonspherical nanoparticles by reducing silver neodecanoate extract with benzyl alcohol. *Theor. Found. Chem. Eng.* 51, 557–562 (2017). The structures, names, molecular weights and solubility parameters of select precursors are found in Supplementary Table 1. The polymerizing components of the resins are described in Supplementary Table 2.*

11) the initiator in this work, TPO-L, should be listed in the experimental section. Currently it can only be found in the SI, the experimental section simply says “initiator”.

TPO-L is now listed in the experimental section on page 14 in the revised manuscript.

12) The paper described the length of the simulated monomers as ‘L = 3, 6, 9’. Figure 4 says N = 3,6,9.

Changes are made in Figure 4 a) and in the Figure caption on page 36 to describe the length of the simulated monomers as ‘L = 3, 6, 9’ in the revised manuscript.

New text on page 36:

Figure 4. a) Diffusion coefficient of a probe molecule in crosslinkers of different spacer lengths ($L=3, 6$ and 9) before and after the formation of a polymer network. The inset shows the diffusion coefficient of the probe molecule (i.e. a molecule 3 beads long), representing silver neodecanoate as a function of the crosslinker spacer length in the formed network. Snapshots from the simulations of the probe molecule (green molecule) in a network formed with crosslinker b) $L=3$ and c) $L=9$.

13) It is very difficult to tell what is happening in Figure 4b,c. Clearly there is a molecular (green balls) trapped in a network, but beyond that what is Figure 4b,c trying to communicate?

We modified Figure 4b and c to more clearly highlight the polymer network. The purpose of Figure 4b and c is to show the size of the probe molecule relative to the crosslinking density of the network. The distance between crosslinking points for the $L=9$ network is greater than that of to the $L=3$ polymer network. It is the looser network of the $L=9$ polymer that allows the probe molecule to more freely diffuse in the polymer network in comparison to the $L=3$ polymer network.

The new Figure and captions on page 36:

Figure 4. a) Diffusion coefficient of a probe molecule in crosslinkers of different spacer lengths ($L=3, 6$ and 9) before and after the formation of a polymer network. The inset shows the diffusion coefficient of the probe molecule (i.e. a molecule 3 beads long), representing silver neodecanoate as a function of the crosslinker spacer length in the formed network. Snapshots from the simulations of the probe molecule (green molecule) in a network formed with crosslinker b) $L=3$ and c) $L=9$. The red balls represent crosslinking points while the blue parts are the bridging segments of the crosslinker. The high density of crosslinking points of the $L=3$ system creates a tight polymer network that impedes diffusion of the probe molecule. In

comparison, the L=9 system allows for less constrained diffusion of the probe molecule due to its lower density of crosslinking points and longer segments between crosslinking nodes.

14) On page 17, the authors write: “This density is high enough to approximate the pure polymer melt of the experiments, but is also low enough to allow for the movement of individual crosslinker molecules on simulation time scales.” What is meant by “high enough and low enough”, please cite relevant sources.

We agree with the reviewer, that the description needs to be more quantitative. A system consisting of Lennard-Jones spheres with no attraction, which is essentially what the current system is, will begin to crystallize at a volume fraction of 0.492.[Dyre, J. C. Simple liquids’ quasiuniversality and the hard-sphere paradigm. J. Phys. Condens. Matter 28, 323001 (2016)] Hence, in order to maintain liquid-like behaviour, the volume fraction must not be higher than this. However, the volume fraction should also not be much lower than this value as otherwise voids of empty space could appear as the system begins to polymerize. This tuning was verified by preliminary simulations in which several volume fractions were tested.

This section of the paper was been rewritten to reflect this quantitative justification:

Old text:

The box length was set to 20σ and each system was filled with enough polymers to achieve a volume fraction of $\sim 49\%$. This density is high enough to approximate the pure polymer melt of the experiments, but is also low enough to allow for the movement of individual crosslinker molecules on simulation time scales. This middle ground then mimics the experimental setup while permitting the exploration of the dynamics of the system.

New text on page 20:

The box length was set to 20σ and each system was filled with enough polymers to achieve a volume fraction of 0.491. Simulations using the same particle model have demonstrated that the system will begin to crystallize at a volume fraction of 0.492.³⁹ Setting the volume fraction to 0.491, thus, mimics the experimental setup by retaining liquid-like behaviour while ensuring that no voids will form during the polymerization process.

15) in the section entitled “3D PIPS smart objects” – the authors describe a truss structure, is this the structure shown in Figure 1C? A image of the truss structure would help clarify the SEM images in Figure 5.

Details of the truss have been included in the Supplementary Figure 9 (see answer to question 17).

16) The authors also write, “formulations yield thermosets with low stiffness”. Please clarify what is meant by “low stiffness”. Please provide compression data or other evidence.

We measured the strain-stress curves of a select few samples; however, we did not do an exhaustive study due to the number of samples prepared in this study. The data was acquired using resins containing 25 wt % crosslinker and DA-250, DA-575 and DA-700 crosslinkers.

As a control, samples containing no silver were measured (blue data in chart) and compared to the samples containing silver (orange data in chart). The data shows that the elastic modulus decreases slightly with the incorporation of the silver. The elastic modulus also decreases with the length of the bridging group of the crosslinker, as would be expected. We will include this data in the SI along with the experimental details used to acquire the data.

New text on page 11:

With the printed objects being compressible (see Supplementary Figure 8 for selected elastic modulus of cylinders), upon applying pressure, the polymer matrix deforms creating new contacts between the silver domains, increasing the conduction pathway and, thus, decreasing resistance (Figure 5a).

New Supplementary Figure on page 14:

Supplementary Figure 8. *The elastic modulus of the various polymers using 25% crosslinker. With increasing crosslinker MW, the elastic modulus decreases. Elastic modulus was determined from tensile tests on samples approximately 1.5 mm in diameter and 1 cm in length. The test instrument, a LEX820/FDAS770 automated fiber tensile test system from Dia-Stron (Andover, UK), measured the sample cross-section using an on-board laser micrometer followed by tensile testing using a high resolution extensometer with 20 N load cell. Four samples of each type were measured to report the average result and standard deviation.*

17) In general, using stress-sensors that operate on electrical conductivity of nanoparticles (graphene, metals, etc.) impeded in a polymer matrix is well known and widely utilized strategy. This section could benefit from further discussion, specifically in the context of sensitivity of their devices in comparison to other published work. A quick google search turns up numerous reviews, including doi: 10.1186/s11671-019-3084-x.

To address this point, we have re-written segments of the manuscript and have elaborated on it in the Supplementary Information as follows:

New text in the manuscript on page 11-12:

Using purposefully formulated resins to control the placement of AgND, we demonstrate the value in being able to tune the surface morphology of printed objects with particular functional properties by considering three applications: strain sensors, antennas and antimicrobial objects. The 3D PIPS approach provides the ability to generate strain sensors that combine complex 3D geometries with piezoresistive properties and holds promise in wearable electronics and motion sensing.⁴² Truss structures were 3D printed using a resin formulation that yields graded silver compositions. The silver is sufficiently concentrated at the surface to form a percolated path for electrical conduction; however, polymer inclusions within the surface silver layer introduces barriers to conduction. With the printed objects being compressible (see Supplementary Figure 8 for selected elastic modulus of cylinders), upon applying pressure, the polymer matrix deforms creating new contacts between the silver domains, increasing the conduction pathways and, thus, decreasing resistance (Figure 5a). By controlling how the silver salt migrates, we can modulate the density of silver at the surface and, thus, its electrical response to compression. The SEM images of the surfaces and cross-sections of strain sensors of Figure 5b, made using 39 and 58 wt % crosslinker, demonstrate the differences in the amount of silver present at the surface. For the truss made with 39 wt % crosslinker, the surface features a dense film of Ag nanoparticles with low electrical resistance whereas the truss made with 58 wt % crosslinker has a surface morphology with sparser particles and, correspondingly, higher electrical resistance (Supplementary Figure 9). The relative decrease in resistance depends on the extent to which the conduction pathway is hindered by the presence of polymer at the surface. Thus, trusses made with high crosslinker concentrations have silver coatings with a more obstructed conduction pathway in comparison to trusses made with less crosslinker. Therefore, these trusses will create a greater number of new silver-to-silver contacts during compression resulting in a greater change in resistance. As shown in Figure 5c, the change in resistance increases with the wt % crosslinker used to make the trusses. The benefit of this approach is that as opposed to varying the filler loading to tune the sensor response, here 3D printed piezoresistive sensors can be made to respond with a given electrical response by simply controlling the phase separation of silver through the resin formulation. In addition, segregated silver at the interface allows one to reach a percolation threshold with a lower loading of conductive filler in comparison to commonly used conductive composite morphologies⁴³ making more efficient use of the conductive material, improving the electrical conductance of the silver phase and minimizing impact on the mechanical properties of the bulk object polymer phase. The strain sensors were found to have gauge factors (i.e. ratio of relative change in electrical resistance to the mechanical strain) of 2.3, 3.2, 5.1 and 15.7 for the sensors made with 39, 46, 52 and 58 wt % crosslinkers as shown in Figure 5d, similar to those reported for 2D strain sensors.⁴⁴ This example illustrates how gauge factors can be dialed-in by simply varying the crosslinker concentration enabling one to target a sensitivity regime of 3D printed piezoresistive sensors.

New figure and caption on Page 37-38:

Figure 5. Strain sensor measurements. a) Schematic representation of the compression mechanism within a truss. The greater the wt % of crosslinker used, the more the conduction channel is obstructed by the polymer phase. With an applied pressure, the polymer phase deforms creating new silver-to-silver connections and decreasing the resistance. The trusses that have highly impeded conduction channels due to a high polymer fraction at its surface create a higher number of new silver-to-silver connections upon compression than the truss with less polymer and, thus, respond accordingly with a greater decrease in resistance. b) SEM images of the top surface

and cross-section of truss structures prepared using different wt % crosslinker. c) Change in resistance upon compression cycling of truss structures prepared with different wt % crosslinker. Trusses made with 39, 46 and 52 wt % crosslinker responded to a compression of 250 μm with maximum changes in resistances of 2.5, 6.0 and 12.5 %, respectively. d) Gauge factors of the trusses as a function of the concentration of crosslinker used in the resin. The gauge factor increases with increasing crosslinker concentration illustrating the strength of this approach in being able to target a given piezoresistive sensor sensitivity through the formulation of the resin.

New Supplementary text and Figure on page 15:

The trusses, with the design shown in Supplementary Figure 9a and dimensions of 11.24 x 11.24 x 13.40 mm, were 3D printed using resins containing various crosslinker concentrations. Compression cycling of the truss structures showed trusses made with 39, 46, 52 and 58 wt % crosslinker responded to a 250 μm compression with maximum changes in resistance of 2.5, 6.0, 12.5, and 48 % respectively.

Supplementary Figure 9. a) Isometric, top and side views of the truss structure used in the strain sensor experiments. b) Photograph of 3D printed truss with dimensions of 11.24 x 11.24 x 13.40 mm. c) Relative resistance of trusses under compression for trusses made with resins containing various crosslinking concentrations.

New text in the Experimental Section on page 22:

The gauge factor was calculated as follows:

$$GF = \frac{\left[\frac{\Delta R}{R_0}\right]}{\left[\frac{\Delta L}{L_0}\right]}$$

where ΔR is the difference in resistance between zero compression (R_0) and a compression of 250 μm (R), L_0 is the length of the truss (13.4 mm) and ΔL is the change in length (250 μm).

18) the image in Figure 6B is too small to read. Please provide a scale bar for the picture in 6A.

Figure 6B is made readable and dimensions are provided for the picture in 6A in the revised manuscript on page 39.

19). **In the experimental section, the authors write: “These resin formulations are comprised of 8.0 wt % of silver metal or 9.5 wt % silver metal post-sintering assuming neodecanoate and 2-ethyl-oxazoline become volatile during the process.” This could be confirmed via thermal gravimetric analysis or simply weighing the sample after sintering.**

The thermogravimetric analysis of 3D printed and sintered cylinders in air with and without Ag are added in the revised text as Supplementary Figure 2 and text is revised in the experimental section on page 15 in the revised manuscript. See the underlined text below.

New Table, Figure and text in the supplementary information on page 6-8:

2. Silver content and spatial distribution of silver

The nominal amount of silver present in the 3D printed cylinders is described in Supplementary Table 3 in terms of the weight fraction of silver neodecanoate in the formulation, the weight fraction of silver in the formulation and the amount of silver in the sintered sample, assuming 2-ethyl-2-oxazoline and the neodecanoate is given off during sintering. The amount of silver in the printed object in comparison to the calculated amount of silver in the formulation was confirmed using thermogravimetric analysis (TGA) of cylinders containing 25 wt % AgND (or 9.5 wt % silver post-sintering), as shown in Supplementary Figure 2a and b. The residual amount of silver in the control (no silver) and in a sintered cylinder is -0.03 and 9.66 wt %, respectively, in agreement with the nominal amount of silver. The amount of silver neodecanoate used throughout the manuscript was 25 wt % AgND as highlighted in bold in Supplementary Table 3.

The amount of silver loading was optimized based on the resistance of the cylinders. As shown in Supplementary Table 3, measurable electrical conductivity was observed for resins containing >19 wt % AgND. The resistance of the surface of the cylinders is at a minimum and does not change significantly when the concentration of AgND in the resin was between 25 and 38 wt %

while AgND amounts greater than 38 wt % reduced the printing ability and strength of the objects. Optical microscope images of the cylinders printed using 25, 30 and 38 wt % AgND are shown in Supplementary Figure 1.

Supplementary Figure 1. Optical microscope images of cylinder printed using 25, 30 and 38 wt % AgND and using resins containing 35 wt % crosslinker DA-250.

Supplementary Table 2. Resistance as a function of different wt % of silver precursor (AgND, silver neodecanoate and 2-ethyl-2-oxazoline) in 35 wt % crosslinker DA-250 formulations and their sintered samples.

wt % AgND**** precursor in formulation*	wt % Ag in formulation**	wt % Ag in sintered sample***	Resistance per cm (Ω)	Sheet resistance Ω/\square
6.25	1.97	2.38	Not conducting	-
12.50	3.94	4.75	Not conducting	-
18.75	5.91	7.16	20.0 ± 5	6.3
25.00	7.88	9.50	1.3 ± 0.1	0.4
30.11	9.49	11.88	0.9 ± 0.1	0.3
37.50	11.82	14.25	3.6 ± 0.7	1.1
50.00	15.76	19.00	Poor electrical and mechanical properties	-

*pre-sintering silver neodecanoate content (column 1):

$$\text{wt \% AgND} = \text{wt AgND} / [\text{wt AgND} + \text{wt resin}] \times 100\%$$

**pre-sintering silver content (column 2):

$$\text{wt \% Ag} = \text{wt Ag} / [\text{wt AgND} + \text{wt resin}] \times 100\%$$

***post-sintering silver content (column 3):

$$\text{wt \% Ag} = \text{wt Ag} / [\text{wt Ag} + \text{wt resin}] \times 100\%$$

****AgND defined as silver neodecanoate with 2-ethyl-2-oxazoline in weight ratios 1:0.22.

Supplementary Figure 2. TGA of 3D printed and sintered cylinders in air. Cylinders are printed (a) without and (b) with 25 wt % of AgND in the resin.

New text on page 15:

These resin formulations are comprised of 8.0 wt % of silver metal or 9.5 wt % silver metal post-sintering assuming neodecanoate and 2-ethyl-oxazoline become volatile during the process (see TGA in Supplementary Figure 2). The amount of silver in the printed object in comparison to the calculated amount of silver in the formulation was confirmed using TGA of cylinders containing 25 wt % AgND (or 9.5 wt % silver post-sintering) as shown in Supplementary Figure 2a and b.

Reviewer #3 (Remarks to the Author):

Deore et al report a method to 3D print multi-material silver composites that utilize phase separation to create different spatial distributions of silver to afford different morphologies. The fabrication of multi-material constructs via 3D printing represents a significant challenge to the field. In particular, there are very few strategies for multi-material vat photopolymerization that have been developed, which the authors appropriately cited. In this manuscript, the authors propose a method for making silver composites with different morphologies based on polymerization-induced phase separation. The authors propose that the silver migration through the matrix material is dependent upon the kinetics of gelation, polymer network density, and material diffusivity.

While the concept is interesting, this manuscript does not meet my expectations for advancements in the field reported in this journal. Moreover, the authors left many unanswered questions that are important to supporting the claims made in this manuscript.

The authors have tried to answer all unanswered questions that are important to supporting the claims made in this manuscript by providing detailed explanation and supporting data in the revised text.

1) Polymerization induced phase separation (PIPS) typically involves phase separation as a result of polymer chain growth, and this does not appear to be the case in the work. In the case of this manuscript, the samples are sintered at 210 C to convert AgND precursors into metallic silver. The authors did not provide clear evidence that AgND is phase separated prior to sintering. So how do the authors know that their observations are not due to different kinetics of polymer degradation? The different compositions of the resin could afford different rates of degradation, which could also affect the phase separation that is observed. Along the same lines, what was the mass loss after sintering?

The authors agree with the reviewer's point that polymerization induced phase separation (PIPS) typically involves phase separation as a result of polymer chain growth. We explained the 3D PIPS mechanism during vat polymerization through schematics in Figure 1 and related text in the section 'Crosslinkers drive the spatial distribution of silver' from pages 4 to 8. To confirm that the silver phase separates during printing, we have included SEM micrographs and EDS maps of pre-sintered samples in Figure 2a and Supplementary Figure 3. The SEM images and EDS analysis proves there is enrichment of the silver salt at the surface of the cylinder prior to sintering. To emphasize this point, we moved some of these SEM images from the supplementary information section into the main manuscript (Figure 2a).

We do not suspect that polymer degradation is significant given that a TGA of the sintered cylinder yields a residual mass similar to what is expected based on the nominal amount of organics material and silver. That is, the cylinders contain a nominal amount of 9.5 % silver post-sintering (see added Supplementary Table 3 below). The TGA of the sintered cylinders

yields a residual mass of 9.6 wt % (see new Supplementary Figure 2). In addition to this TGA, we provide a TGA of the polymer heated at the same temperature used to sinter the polymer. The TGA shows that the mass loss of the polymer is not significant enough to cause the silver phases to form as a result of polymer degradation. We have added this data in the Supplementary Information (Supplementary Figure 15) on page 23.

New text on page 15: The amount of silver in the printed object in comparison to the calculated amount of silver in the formulation was confirmed using TGA of cylinders containing 25 wt % AgND (or 9.5 wt % silver post-sintering) as shown in Supplementary Figure 2a and b.

New Table and Figures in the supplementary information on page 6, 8 and 23:

Supplementary Table 3. Resistance as a function of different wt % of silver precursor (AgND, silver neodecanoate and 2-ethyl-2-oxazoline) in 35 wt % crosslinker DA-250 formulations and their sintered samples.

wt % AgND**** precursor in formulation*	wt % Ag in formulation**	wt % Ag in sintered sample***	Resistance per cm (Ω)	Sheet resistance Ω/\square
6.25	1.97	2.38	Not conducting	-
12.50	3.94	4.75	Not conducting	-
18.75	5.91	7.16	20.0 \pm 5	6.3
25.00	7.88	9.50	1.3 \pm 0.1	0.4
30.11	9.49	11.88	0.9 \pm 0.1	0.3
37.50	11.82	14.25	3.6 \pm 0.7	1.1
50.00	15.76	19.00	Poor electrical and mechanical properties	-

*pre-sintering silver neodecanoate content (column 1):

$$wt \% AgND = wt AgND / [wt AgND + wt resin] \times 100\%$$

**pre-sintering silver content (column 2):

$$wt \% Ag = wt Ag / [wt AgND + wt resin] \times 100\%$$

***post-sintering silver content (column 3):

$$wt \% Ag = wt Ag / [wt Ag + wt resin] \times 100\%$$

****AgND defined as silver neodecanoate with 2-ethyl-2-oxazoline in weight ratios 1:0.22.

Supplementary Figure 2. TGA of 3D printed and sintered cylinders in air. Cylinders are printed (a) without and (b) with 25 wt % of AgND in the resin.

Supplementary Figure 15. TGA of pre-sintered 3D printed cylinders using a resin mixture (75 mL of 50 wt % DA-575 with 25 mL of 35 wt % DA-250) that contained no silver and treated with 5 minutes of UV curing. The mass loss of the cylinders were analyzed by TGA under isothermal condition at 210°C (i.e. sintering temperature). The mass loss after 1 h is 1.4% demonstrating that the polymer does not degrade significantly during the sintering step.

New text on page 17:

The mass loss of the 3D printed cylinders that contained no silver were analyzed by TGA under isothermal condition at 210°C. The mass loss after 1 h is minimal demonstrating that the polymer does not degrade significantly during the sintering step (See Supplementary Figure 15).

2) Why are there not nano- and micro-islands of silver metal within the polymer network? It is unclear why silver would migrate to the air interface.

There does exist inclusions of silver metal within the polymer network. As described in the manuscript, the extent of inclusions is dependent on the formulation of the resin. Resins that impede diffusion of the silver salt tend to have more inclusion of silver within its core while those that allow the silver salt to migrate freely during polymerization have less silver salt in the bulk of the object. This effect is represented in Figure 2b of the manuscript and Supplementary Figure 4 that show the spatial composition of the cylinders with a small amount of silver present in the core of the object. The morphology of these inclusions can be visualized in the SEM images of Figure 2b, as well as in the SEM images of the antimicrobial samples prepared with 0.5 and 1.0 wt % of Ag in Supplementary Figure 11 and strain sensor SEM images in Figure 5b.

An additional SEM image of the core of the cylinder has been included in Supplementary Figure 4 on page 10 to highlight this point.

The silver does not migrate to an air interface, but to the interface between the printed object and the resin during the printing process. The silver salt migrates towards the resin phase due to strong entropic driving forces, with the resin providing a more favorable mixing environment than the polymer phase.

We have included an SEM image in Supplementary Figure 4 to show silver inclusions in the center of a 99 wt % DA-575 cylinder.

New Supplementary Figure 4 and caption on page 10:

Supplementary Figure 4. a) The Ag wt % as a function of distance away from the surface along the cross-section of a 1.5 mm diameter cylinder. The center of the cylinders have as low as 4 wt % Ag while those made with 99% crosslinker have silver concentrations with 15-20 wt % within its core. b) SEM image of the center of 99 wt % DA-575 showing inclusions of silver.

3) Pg 5, ln 22: The authors should define exactly what these percentages represent.

The percentages on Pg 6, ln 22 are defined and referred to the Supplementary Table 2 in the revised text.

New text on page 5: The four resin systems, distinguished by the length of the PEG spacer of the diacrylates, formed the basis of this study (170, 250, 575 and 700 g/mol M_n PEG-diacrylates are referred to as DA-170, DA-250, DA-575 and DA-700, respectively; Supplementary Table 2).

4) **The authors mention a “probe molecule” on page 9 but never stated the identity of the compound.**

The definition of the probe molecule is defined in the text, but we have included a definition in caption of Figure 4.

5) **Pg 9: The authors discuss how shorter crosslinkers afford higher density of crosslinking, which leads to smaller pore size. The authors should clarify whether they are referring to pore size or mesh size.**

We are referring to the length between crosslinking points when we refer to the pore size. We modified the language as follows to clarify.

New text is underlined and found on page 9-10:

The diffusivity of AgND will impact the amount of Ag that accumulates at the surface and may explain the observed differences in surface silver for a given delay time. The diffusivity of AgND will change during polymerization as a result of increases in viscosity and constraints imparted by the growing polymer network.⁴¹ The extent to which the diffusivity of AgND will change when the resin is transformed into a polymer network will be highly dependent on the length of the spacer between reactive moieties in the crosslinker or, in other words, the crosslinking density. To explore this idea, coarse-grained Langevin dynamics simulations of a simplified system were performed (see Methods for details). The simulations tracked the displacement of a probe molecule, representing silver neodecanoate, in systems containing 100 wt % crosslinkers. The simulations monitored the diffusivity of the probe molecule in unreacted resin and in a fully formed polymer network only, and therefore, the simulations did not require a consideration of kinetic effects of the polymerization reaction.

The diffusion coefficient of the probe molecule in crosslinkers of different lengths (L=3, 6 and 9) is shown in Figure 4a. In the absence of any polymerization, the diffusion coefficient is higher in the short crosslinker. This is to be expected as the viscosity of the solution increases with increasing MW. However, for the case of diffusion in the polymer networks, the diffusion coefficient is highest for the longest crosslinker. In this limit, the networks formed by the shorter crosslinkers have a higher density of crosslinking points and, correspondingly, a tighter network for the probe molecule to travel through than the longer crosslinkers. This can be seen by examining the images for the probe molecule in the L=3 network (Figure 4b) and in the L=9 network (Figure 4c). The length of the linear bridging segments (blue segments of Figure 4b and c) and the density of the crosslinking points (red segments) that define the density of the polymer network are distinctly different in the L=3 and L=9 network. The shorter length between crosslinking points of the L=3 network causes the diffusion of the probe molecule to be more constrained in comparison to the L= 9 network. Note that the y-axis in Figure 4a is logarithmic and, thus, the decrease in diffusivity for the short crosslinkers is much more dramatic than for the long crosslinkers. The diffusion of the probe molecule is ~24 times greater in the unreacted resin than in the network for L=3, but it is only ~2 times greater for L=9.

We also modified Figure 4 and its caption (see answer 13 to referee 2).

6) The explanation of how the strain sensor works on page 11 was unclear, particularly in reference to Figure 5b.

We modified the text and the Figure captions as follows on page 11-12:

Using purposefully formulated resins to control the placement of AgND, we demonstrate the value in being able to tune the surface morphology of printed objects with particular functional properties by considering three applications: strain sensors, antennas and antimicrobial objects. The 3D PIPS approach provides the ability to generate strain sensors that combine complex 3D geometries with piezoresistive properties and holds promise in wearable electronics and motion sensing.⁴² Truss structures were 3D printed using a resin formulation that yields graded silver compositions. The silver is sufficiently concentrated at the surface to form a percolated path for electrical conduction; however, polymer inclusions within the surface silver layer introduces barriers to conduction. With the printed objects being compressible (see Supplementary Figure 8 for selected elastic modulus of cylinders), upon applying pressure, the polymer matrix deforms creating new contacts between the silver domains, increasing the conduction pathways and, thus, decreasing resistance (Figure 5a). By controlling how the silver salt migrates, we can modulate the density of silver at the surface and, thus, its electrical response to compression. The SEM images of the surfaces and cross-sections of strain sensors of Figure 5b, made using 39 and 58 wt % crosslinker, demonstrate the differences in the amount of silver present at the surface. For the truss made with 39 wt % crosslinker, the surface features a dense film of Ag nanoparticles with low electrical resistance whereas the truss made with 58 wt % crosslinker has a surface morphology with sparser particles and, correspondingly, higher electrical resistance (Supplementary Figure 9). The relative decrease in resistance depends on the extent to which the conduction pathway is hindered by the presence of polymer at the surface. Thus, trusses made with high crosslinker concentrations have silver coatings with a more obstructed conduction pathway in comparison to trusses made with less crosslinker. Therefore, these trusses will create a greater number of new silver-to-silver contacts during compression resulting in a greater change in resistance. As shown in Figure 5c, the change in resistance increases with the wt % crosslinker used to make the trusses. The benefit of this approach is that as opposed to varying the filler loading to tune the sensor response, here 3D printed piezoresistive sensors can be made to respond with a given electrical response by simply controlling the phase separation of silver through the resin formulation. In addition, segregated silver at the interface allows one to reach a percolation threshold with a lower loading of conductive filler in comparison to commonly used conductive composite morphologies⁴³ making more efficient use of the conductive material, improving the electrical conductance of the silver phase and minimizing impact on the mechanical properties of the bulk object polymer phase. The strain sensors were found to have gauge factors (i.e. ratio of relative change in electrical resistance to the mechanical strain) of 2.3, 3.2, 5.1 and 15.7 for the sensors made with 39, 46, 52 and 58 wt % crosslinkers as shown in Figure 5d, similar to those reported for 2D strain sensors.⁴⁴ This example illustrates how gauge factors can be dialed-in by simply varying the crosslinker concentration enabling one to target a sensitivity regime for 3D printed piezoresistive sensors.

New figure and caption on Page 37-38:

Figure 5. Strain sensor measurements. a) Schematic representation of the compression mechanism within a truss. The greater the wt % of crosslinker used, the greater the conduction channel is obstructed by the polymer phase. With an applied pressure, the polymer phase deforms creating new silver-to-silver connections and decreasing the resistance. The trusses that have highly impeded conduction channels due to a high polymer fraction at its surface create a higher number of new silver-to-silver connections upon compression than the truss with less polymer and, thus, respond accordingly with a greater decrease in resistance. b) SEM images of the top surface and cross-section of truss structures prepared using different wt % crosslinker. c) Change in

resistance upon compression cycling of truss structures prepared with different wt % crosslinker. Trusses made with 39, 46 and 52 wt % crosslinker responded to a compression of 250 μm with maximum changes in resistances of 2.5, 6.0 and 12.5 %, respectively. d) Gauge factors of the trusses as a function of the concentration of crosslinker used in the resin. The gauge factor increases with increasing crosslinker concentration illustrating the strength of this approach in being able to target a given piezoresistive sensor sensitivity through the formulation of the resin.

7) Some control experiments for the strain sensor, antennae, and antimicrobial surface would have improved the impact of these experiments.

The control experiment for the **antimicrobial surface** were provided in Supplementary Figure 12. The control in this case is that of a printed object containing no silver. The control shows lower antimicrobial activity than the printed objects with silver.

We included a control measurement for the **antenna** in Supplementary Figure 10.

New text on page 13:

The gain measurement performed in an anechoic chamber using a gain standard horn antenna, as shown in Supplementary Figure 10, are comparable with the literature reports.⁴⁵

New text and Figure in Supplementary Information on page 16-17:

*The measurement of gain was performed in an anechoic chamber using a gain standard horn antenna as shown in Supplementary Figure 10a below. The gain standard was used as the receive antenna (antenna array in Figure 6) and *s*-parameter measurements collected as they were for the antenna array with the VNA. With a peak measured *s*-parameter magnitude of -54.8 dB and a known maximum gain for the standard of 7.8 dB gives a conversion of*

$$G = S_{21} + 62.6 \text{ dB} \quad (1)$$

*where G is gain and S_{21} is the magnitude of transmission, both in dB. This equation allows the *s*-parameter data to be converted to gain values assuming negligible return loss. In this case both the gain standard and the printed dipole array were matched to 50 Ω at 2.4 GHz with return loss values better than -10 dB.*

*Making the *s*-parameter measurements of the antenna array and converting to gain using Equation 1 gives the plot as shown in Supplementary Figure 10b. This measured data is shown plotted beside simulated data from Ansys HFSS.*

In HFSS, the array model (shown in Supplementary Figure 10c) used a finite surface impedance for all conductors of $1 \times 10^6 \text{ S/m}$. From Supplementary Figure 10b, the simulated results show 4.1 dB more gain than the measured results. This discrepancy in results could be due to losses from the conductive adhesive used to join the dipole antenna branches to the conductor or the PCB header pins used as through hole vias to connect the dipoles through to the ground plane. It could also be due to a breakage that happened in the middle of one antenna during measurement. The break was repaired with copper tape and checked for DC conductivity, but its

gain could have been reduced by the repair. This could also explain the Asymmetry in the measured radiation pattern with a significant side lobe at an angle of $\theta = 90^\circ$.

Other dipole antennas have been 3D printed such as shown in [1] where the design was metalized with copper air brushing and set a quarter wavelength above a ground plane for enhanced radiation. The maximum reported gain for the dipole in [1] was 6.5 dB. To close the gap between this number and the 1.3 dB reported here, the dipoles could be optimized for interaction with a ground plane by elevation to a quarter wavelength for enhanced gain from the ground plane reflection. Thicker vias could create less inductance for a stronger connection between the ground plane and the base of the dipole. A high conductivity adhesive such as copper tape could be used instead of conductive epoxy to reduce transition losses at the antenna. Adding a balun at the input to each dipole could also reduce transition losses and improve the overall gain of the array.

[1] Ranjbar Naeini, M., Mirmozafari, M. & Van Der Weide, D. Monolithic 3-D Printing of an Integrated Marchand Balun with a Dipole Antenna. *IEEE Trans. Components, Packag. Manuf. Technol.* 10, 654–658 (2020).

Supplementary Figure 10. (a) Gain standard horn antenna (BAE Systems H-1498). (b) Measured antenna array gain compared with HFSS simulation. (c) HFSS model of dipole antenna array with feeding network and metal conductivity of 1×10^6 Siemens/m.

Control experiments are difficult to define for **strain sensor** and are not generally used in literature. For that reason, we chose instead to provide gauge factors for our sensors. Gauge factors are commonly reported and allow a comparison with other reported strain sensors. The text of the manuscript has been modified. See answer to question 6.

8) Figure 1 is unclear to me. It is difficult to tell what we are supposed to see, particularly in b. The images in c should have a scale bar or reference.

Figure 1 describes the phase separation process of AgND by PIPS during 3D printing and the ability to tune the final product to either a silver composite, silver gradient or silver coating. Figure 1a depicts the process of an SLA printer and the different phases during printing including the formation of the 3D polymer object, the interface containing phase-separated AgND and oligomers and the remaining bulk resin containing a homogeneous mixture of monomer, crosslinker, photoinitiator and silver precursor. We have adapted Figure 1b to provide more clarity in the three types of functional products (silver composite, silver gradient or silver coating) by moving the conditions and titles above the graphics and SEM images. We have also replaced the SEM images with ones that more distinctly show the three conditions. The dimensions of the lattices in Figure 1c were added to the caption to provide a reference to the scale. The following is the updated Figure 1 and caption on page 31-32:

a)
POLYMERIZATION INDUCED PHASE SEPARATION (PIPS) IN 3D VAT POLYMERIZATION

b)

High crosslinking density
 Rapid gelation

Low crosslinking density
 Slow gelation

c)

Figure 1. 3D PIPS printing mechanism of functional objects. a) Schematic of the phase separation of AgND induced by PIPS (left) depicting the AgND functional material phase separating from the polymer object to the interface during 3D printing using an SLA printer (right). b) The tuning of phase separation is controlled with crosslinker molecular weight and percent crosslinker to alter the crosslinking density and polymer gelation rate to yield different morphologies in 3D objects. Representative SEM images are presented to depict the composite, gradient or coating morphologies. c) Photographs of PIPS induced 3D printed and sintered Ag coated lattice objects with dimensions of 40 x 40 x 40 mm.

9) **Figure 6 should use a larger font size. Also the image in 6a is difficult to see. Was this 3D printed?**

Yes, dipole antennas were 3D printed and then glued on the substrate with conducting epoxy. Details are added in the Experimental Section on page 22. As per the reviewer's suggestion, changes are made in Figure 6 in the revised manuscript on page number 39.

Figure 6. 3D printed dipole antenna array. a) Photograph of antenna array (dimensions 10 cm by 10 cm). b) Functional anechoic chamber layout showing transmission between a gain standard horn antenna and antenna array under test. c) 3D printed dipole antenna array normalized radiation pattern at 2.4 GHz compared with ideal array factor.

10) In Supplementary Figure 3 needs information about the scale bar and a sample of SEM image and EDS layer image cannot read. The SEM image of 1.0 wt % Ag sample is missing, in Supplementary Figure 8. It is hard to see the blue fluorescence regions of EDS images, f, and g in Supplementary Figure 10.

We included new scale bars in Supplementary Figure 3 in the revised Supplementary Information on page 9.

Supplementary Figure 3. SEM images, EDS layered electron images and EDS mapping of silver, carbon and oxygen elements of the cross-sections of pre-sintered cylinders made from two different wt % DA-250 resin compositions. Cylinders were dried for two days in fume hood at room temperature.

We included SEM images of the 0.5 wt % Ag sample to show the incomplete coverage of silver on the surface. We included the SEM image of the 1.0 wt % Ag in Supplementary Figure 11 on page 18.

Supplementary Figure 1. An SEM image of the surface of 3D printed the anti-microbial sample made from resins containing a) 0.5 wt % and b) 1.0 wt % AgND.

The x-ray emission used to map the elemental composition of samples using EDS has inherently low signal to noise, unlike other molecular fluorescence imaging techniques. We have, however, imaged with longer acquisition times to improve the signal to noise (Supplementary Figure 13 on page 20).

Supplementary Figure 13. (a) Variation of the titanium as a function of distance from the surface of a cylinder printed with TiO_2 nanoparticles and as measured by EDS. (b) SEM image of surface of cylinder printed with TiO_2 nanoparticles and image of printed cylinders (insert). (c) Cross-sectional SEM image of a cylinder printed with TiO_2 nanoparticles. (d) Cross-sectional SEM image of a cylinder printed with barium strontium titanate nanoparticles. (e – f) SEM/EDS cross-section image of a cylinder printed with iron oxide nanoparticles. The nanoparticles appear as bright areas in the SEM. EDS mapping of carbon (f) and iron (g) elements in the sample.

REVIEWERS' COMMENTS

Reviewer #2 (Remarks to the Author):

Deore et al. have made considerable and satisfactory efforts to address the comments and question of the reviewers. Informative and clarifying discussion has been added to the manuscript and several of the figures have been revised for clarity. Moreover, significant information has been added to the supporting information. It is our opinion that this manuscript is now suitable for publication in Nature Communications.

Reviewer #3 (Remarks to the Author):

The authors have provided a revised manuscript that is significantly better than the original, and I appreciate the authors' care to address my concerns. The updated figures, SI figures, and explanations are a vast improvement. This manuscript is suitable for publication.